# Prevalence and incidence of hypertension in a heavily treatment-experienced cohort of people living with HIV in Uganda

Dathan M. Byonanebye[1,2]*, Mark N. Polizzotto[3], Rosalind Parkes-Ratanshi[4,5], Joseph Musaazi[4], Kathy Petoumenos[1], Barbara Castelnuovo[4]

1 Kirby Institute, University of New South Wales, Sydney, Australia, 2 School of Public Health, Makerere University, Kampala, Uganda, 3 The Australian National University, Canberra, Australia, 4 Infectious Diseases Institute, Kampala, Uganda, 5 Cambridge Institute of Public Health, University of Cambridge, Cambridge, United Kingdom

* dbyonanebye@musph.ac.ug

**Data Availability Statement:** Data cannot be shared publicly as it contains potentially identifying participant information. In addition, there are

## Abstract

### Introduction

The effect of long-term exposure to antiretroviral therapy (ART) on hypertension in sub-Saharan Africa remains unclear. We aimed to determine the prevalence and incidence of hypertension in people living with HIV (PLWH) with more than 10 years of ART in Uganda.

### Methods

The analysis was performed within a cohort of adult PLWH with more than 10 years of ART at an HIV clinic in Kampala, Uganda. Participants were eligible for this analysis if they had ≥2 follow-up visits. Hypertension was defined as two consecutive systolic blood pressure (SBP) measures greater than 140 mmHg and/or diastolic blood pressure (DBP) greater than 90 mmHg, and/or documented diagnosis and/or the initiation of antihypertensives. We determined the proportion of PLWH with hypertension at baseline and used multivariable logistic regression to determine the factors associated with prevalent hypertension. To determine the incidence of hypertension, follow-up began from the cohort baseline date and was censored at the last clinic visit or date of the event, whichever occurred earlier. Multivariable Poisson regression was used to determine the adjusted incidence rate ratios (aIRR) of hypertension according to demographic, ART, and clinical characteristics.

### Results

Of the 1000 ALT participants, 970 (97%) had ≥2 follow-up visits, and 237 (24.4%) had hypertension at baseline. The odds of prevalent hypertension were 1.18 for every 5-year increase in age (adjusted odds ratio (aOR) 1.18, 95% CI 1.10–1.34) and were higher among males (aOR 1.70, 95% CI 1.20–2.34), participants with diabetes mellitus (aOR 2.37, 95% CI 1.10–4.01), obesity (aOR 1.99, 95% CI 1.08–3.60), high cholesterol (aOR 1.47, 95% CI 1.16–2.01), and those with prior exposure to stavudine (aOR 2.10, 95% CI 1.35–3.52), or nevirapine (aOR 1.90, 95% CI 1.25–3.01). Of the 733 participants without hypertension at baseline, 116 (15.83%) developed hypertension during 4671.3 person-years of follow-up

restrictions on data sharing imposed by the infectious Diseases Institute and the ethics review committees, in line with the Uganda Data Protection and Privacy Act 2019. Therefore, interested researchers should send data requests jointly to: Barbara Castelnuovo, Head of Department – Research, Infectious Diseases Institute (Mulago Hospital building), P.O. Box 22418, Kampala, Uganda, email: research@idi.co.ug and to The Chair, Joint Clinical Research Council Research Ethics Committee (JCRC REC) Lubowa Hill, Plot 101 Entebbe Road, P. O. Box 10005, Kampala, Uganda. Email: jcrc@jcrc.org.ug.

**Funding:** The ALT cohort has been partly funded through a grant from the Johnson & Johnson corporate citizenship trust. Castelnuovo B is partly funded by the Fogarty International Centre, National Institute of Health (grant# 2D43TW009771-06 "HIV and co-infections in Uganda"). The funders had no role in study design, data collection and analysis, decision to publish, or preparation of the manuscript.

**Competing interests:** The authors have declared that no competing interests exist.

(incidence rate 24.8 per 1000 person-years; 95% CI 20.7–29.8). The factors associated with incident hypertension were obesity (adjusted incidence rate ratio (aIRR) 1.80, 95% CI 1.40–2.81), older age (aIRR 1.12 per 5-year increase in age, 95% CI 1.10,1.25), and renal insufficiency (aIRR1.80, 95% CI 1.40–2.81).

## Conclusion

The prevalence and incidence of hypertension were high in this heavily treated PLWH cohort. Therefore, with increasing ART coverage, HIV programs in SSA should strengthen the screening for hypertension in heavily treated PLWH.

## Introduction

The successful scale-up of antiretroviral therapy (ART) has improved the overall survival of people living with HIV (PLWH) globally and in sub-Saharan Africa (SSA) [1], although survival still lags that of people without HIV [2]. This improvement in survival has contributed to an epidemiological transition in the cause of mortality in PLWH away from AIDS-related opportunistic infections towards non-communicable diseases, including cardiovascular diseases [3, 4]. Globally, the risk of cardiovascular disease is twice as high in PLWH than in people without HIV. In some SSA countries, more than 15% of the cardiovascular burden occurs in PLWH [5]. However, it is unclear whether the increasing cardiovascular burden in PLWH is attributable to ageing and/or the increasing prevalence of cardiovascular disease risk factors, including hypertension. Hypertension is an important independent risk factor for cardiovascular disease, accounting for up to 44% of cardiovascular events in some HIV cohorts [15]. The cardiovascular risk attributable to hypertension is higher in black people than in other ethnicities [6]; however, there is little data on hypertension in heavily treated PLWH in SSA.

Despite increasing ART coverage among PLWH in SSA [7], the impact of long-term ART exposure on cardiovascular risk factors, such as hypertension, has not been fully described [8]. The relationship between ART exposure and hypertension was initially investigated in the Multicenter AIDS Cohort, in which it was demonstrated that exposure to ART for more than two years was associated with a higher risk of hypertension [9]. While this analysis was conducted before ART became widely available, subsequent studies in the modern ART era have also demonstrated a greater risk of hypertension in ART-experienced versus ART-naïve PLWH. A recent systematic review reported higher systolic and diastolic blood pressures of 4.52 mmHg and 3.17 mmHg, respectively, in ART-experienced PLWH than in treatment-naive individuals [10]. The effect of ART on blood pressure appears to be more detrimental in sub-Saharan Africa; a recent study showed that ART for at least three months is associated with an increase in systolic and diastolic blood pressure of 7.85 mm Hg and 7.45 mm Hg, respectively [11]. The prevalence of hypertension among PLWH increased by 10% between 2007 and 2017 [12], possibly due to increasing ART coverage and overall survival.

It is unclear whether the increase in blood pressure following ART initiation is sustained with prolonged exposure to ART and whether the exposure translates to higher rates of hypertension. Additionally, prior analyses have linked older antiretroviral drugs such as nevirapine, indinavir/ritonavir, and stavudine to a higher risk of hypertension [13, 14]. Until recently, these agents were the cornerstones of ART in SSA [15, 16] although it is unclear whether the risk posed by these agents persists in PLWH who switch to contemporary ART regimens, the regimens currently recommended for HIV treatment. Therefore, in this analysis, we sought to

determine the prevalence and incidence of hypertension and its relationship with exposure to specific antiretroviral therapy in a Ugandan cohort of PLWH with long durations of ART.

## Methods

The analysis was conducted within the antiretroviral treatment long-term (ALT) cohort (ClinicalTrials.gov #: NCT02514707), which has previously been described [17, 18]. Briefly, the ALT study is an ongoing single-centre prospective cohort of 1000 PLWH on ART for at least 10 years at baseline. Cohort participants were recruited from the Infectious Diseases Institute (IDI) HIV clinic in Kampala, Uganda, between 2014 and 2015. Because of prolonged exposure to ART at baseline (>10 years), the participants in this cohort were considered heavily treatment-experienced since HIV resistance testing is limited in resource-limited settings. At the cohort baseline, data on HIV treatment, and other covariates, including blood pressure measurements and hypertension treatment, were collected from an existing clinic health information management system. Participants in the cohorts are evaluated annually, blood pressure measurements are taken, and treatment for hypertension and other co-comorbidities, including hypertension, is documented. Blood pressure measurement is standardised at all visits, and readings are taken by qualified nurses using calibrated blood pressure machines, consistent with international guidelines [19].

Data on complications associated with HIV infection and ART is also collected. Laboratory tests (HIV RNA viral load, CD4 count, and creatinine level) are also performed at every visit, whereas lipid tests were only available at baseline. HIV treatment for participants is standardised and follows the World Health Organization (WHO) and Uganda ART guidelines [20, 21]. All PLWH receive a three-drug ART regimen consisting of two nucleos(t)ide reverse transcriptase inhibitors (NRTIs) and non-nucleoside reverse transcriptase inhibitors (NNRTI) as the first-line treatment or protease inhibitors (PIs) as the second-line treatment. In 2018, dolutegravir (DTG) was introduced in Uganda as a part of a three-drug regimen for all patients receiving ART. Patients already on ART were recommended to switch from PIs or NNRTIs to DTG-containing regimens, and the process of switching all participants is underway.

### Participant selection and inclusion criteria into analysis datasets

We included all participants in the cohort with at least two blood pressure measurements recorded at two different follow-up visits. Participants with fewer than two follow-up visits were excluded because hypertension diagnosis requires at least two measurements taken at two different visits [19]. In addition, individuals with hypertension at baseline were excluded from the hypertension incidence analysis. The baseline date for all analyses was the date of enrolment in the cohort (i.e., cohort baseline).

### Definitions and analysis endpoints

Hypertension was defined as systolic blood pressure (SBP) $\geq$ 140 mmHg and/or diastolic blood pressure (DBP) $\geq$ 90 mmHg on two consecutive visits or initiation of antihypertensives including angiotensin-converting enzyme inhibitors (ACEIs). This definition has been used in other analyses of HIV cohorts in SSA [22–24] and a prior analysis of the ALT cohort [18]. The definition is also consistent with international Hypertension diagnosis and management guidelines, which recommend two consecutive abnormal blood pressure measurements [19]. Therefore, prevalent hypertension, determined at baseline, was defined as two consecutive systolic blood pressure (SBP) measures greater than 140 mmHg and/or diastolic blood pressure (DBP) greater than 90 mmHg, and/or documented diagnosis and/or the initiation of antihypertensives within one year before and up to one month after the baseline date. The other

covariates were also defined as consistent with a prior analysis of this cohort [18]. Diabetes mellitus was defined as a clinical diagnosis and/or random blood glucose levels ≥11.1 mmol/L based on point-of-care glucose testing and/or the initiation of antidiabetic medication, consistent with other analyses in HIV cohorts [25, 26]. Hepatitis B infection was defined as a positive hepatitis B surface antigen test result, which is the screening strategy recommended by WHO for high-burden countries [27]. The Chronic Kidney Disease Epidemiology Collaboration (CKD-EPI) equation was used to determine the estimated glomerular filtration rate (eGFR), and renal insufficiency was defined as eGFR <90 mL/min/1.73 m$^2$ [28].

## Statistical analyses

We determined the proportion of PLWH with hypertension at baseline and compared the characteristics of PLWH with hypertension versus those without hypertension. Descriptive statistics for ordinal and categorical variables were computed as frequencies and percentages, and continuous variables were summarised as medians (interquartile ranges, [IQR]). Multivariable logistic regression was used to determine the factors associated with hypertension at baseline. We then performed univariable analysis followed by multivariable Poisson regression to determine hypertension incidence rate ratios (aIRR) and their corresponding 95% confidence intervals (CI) according to exposure to demographic, HIV-infected, and clinical characteristics. While fitting the Poisson regression, the logarithm of person-years of follow-up was included in the model as an offset term to account for the observation time of individuals.

The overall incidence of hypertension was determined and stratified according to demographic, metabolic, and HIV-related factors. Follow-up began from the cohort baseline date and was right censored at the last available study visit or the date of the event, whichever occurred earlier. The factors that were considered in the multivariable model include sex, age, smoking and alcohol status, calendar year, prior AIDS, total cholesterol (TCHOL), triglycerides (TRIG), low-density lipid cholesterol (LDL), body mass index (BMI), diabetes, HIV RNA and CD4 counts, and eGFR and prior exposure to individual antiretroviral drugs (before baseline). These factors were selected a priori based on published data linking them to hypertension [14, 22, 24, 29, 30]. All variables, except age, were modelled as categorical variables, and the categories were selected so that clinically meaningful conclusions could be made. For example, the BMI categories followed the WHO categories [31], and blood pressure categories are based on the European Association for cardiology categorisation for blood pressure [32]. The CD4 categories are also based on the Centers for Disease Control and Prevention (CDC) categories for immunosuppression status [33] while lipids were categorised following the thresholds for cardiovascular risk [34]. All variables in this analysis were fixed at baseline, and the regression models were manually fitted. In the backward selection process, variables with a p-value <0.25 at the univariable stage were considered in the multivariable analysis. All variables dropped from the multivariate model were assessed to determine if they were confounders (i.e., causes >10% change in effects when added in the model), in which case they were retained in the adjusted model. The goodness-of-fit test was used to determine the fitness of the model with the confounder versus the one without the confounder and the one with the smallest Akaike information criterion were considered the better fitting. Specifically, BMI and age have consistently been shown to be confounders for hypertension [14, 22, 24, 29, 30]; their confounding potential was checked, and the variables were consistently included in the model. Variables, other than confounders, that were considered for multivariable analysis but dropped during backward elimination due to non-significance (p<0.05) were later added one at a time to determine adjusted estimates for these variables. We checked for multicollinearity in the variables included in the final model using a variance inflation factor (VIF) cut-off of

5.0. Odds ratios (95% confidence intervals (CI)) and incidence rate ratio (95% CI) were used to determine the association between variables and prevalent and incident hypertension, respectively, and a two-sided p-value <0.05 was considered statistically significant. The final logistic and Poisson regression models were evaluated to ensure compliance with the respective model assumptions. The Wald chi-square test was used to determine whether the final model was statistically significant.

Where data were missing for a variable, an unknown category was assigned, and the variable was fitted as a categorical variable. To determine the impact of missing data and the robustness of estimates, we used the regular Little's test [35], to determine if missing data were missing completely at random (MCAR) and if multiple imputation was necessary. Data for this analysis were prepared using SAS Enterprise Guide software version 8.3 release 5 (SAS Institute Inc., Cary, NC, USA) and analysed with Stata version 16.0 (StataCorp, College Station, Texas, USA). Participants in the ALT cohort provided written informed consent, and the study received ethical approval from the Joint Clinical Research Council Research Ethics Committee (JCRC REC) and Uganda National Council of Science and Technology (UNCST; Approval #: UNCST Folio Number: HS 1586 24th 03 2014).

## Results

### Cohort description and characteristics of study participants

A total of 1000 participants were enrolled in the ALT cohort, 970 of whom had at least two follow-up visits (with at least two blood pressure measurements) and were included in the analysis to determine the prevalence of hypertension (Fig 1). The median number (interquartile range, IQR) of follow-up visits was 5 (4–6), and all visits were annual. The 30 participants who were excluded because they had fewer than two follow-up visits had comparable ages (median age: 45.0; IQR 40.0,51.0) but were mainly male (53%).

### Prevalence of hypertension

Of the 970 participants included in the prevalence analysis, 237 (24.4%) had hypertension at cohort enrolment (baseline). Overall, the median age was 45.4 (IQR 40.4–50.5) years and there were more females (61.8%) than males. A total of 466 participants (48.0%) had experienced an AIDS event before enrolment in the cohort; 960/970 (99.0%) had HIV RNA levels <200 copies/mL, and the median CD4 cell count was 509 (IQR,365–684). The baseline smoking rate was 2.2%. The characteristics of participants with and without hypertension at enrolment are presented in Table 1.

Compared with those without hypertension, participants with hypertension at cohort enrolment were older, more likely to be male, had diabetes mellitus, had higher total cholesterol levels, and had more prolonged exposure to stavudine and nevirapine. After adjustment for confounders, the odds of prevalent hypertension were higher among males (aOR 1.70, 95% CI 1.20–2.34), as well as in participants with diabetes mellitus (2.37, 95% CI 1.10–4.01), obesity (aOR 1.99, 95% CI 1.08–3.60), higher total cholesterol (aOR 1.47, 95% CI 1.16–2.01), and prior exposure to stavudine (aOR 2.10, 95% CI 1.35–3.52), or nevirapine (aOR 1.90, 95% CI 1.25–3.01). In addition, the odds of prevalent hypertension were higher in participants with missing triglyceride levels. Finally, the odds of hypertension were 1.18 (CI 1.10–4.01) for every 5-year increment in the cohort enrolment age (Table 2).

### Incidence of hypertension

After excluding 237 participants with hypertension at enrolment, 733 participants were included in the incidence analysis. Of the 733 included participants, the baseline ART regimen

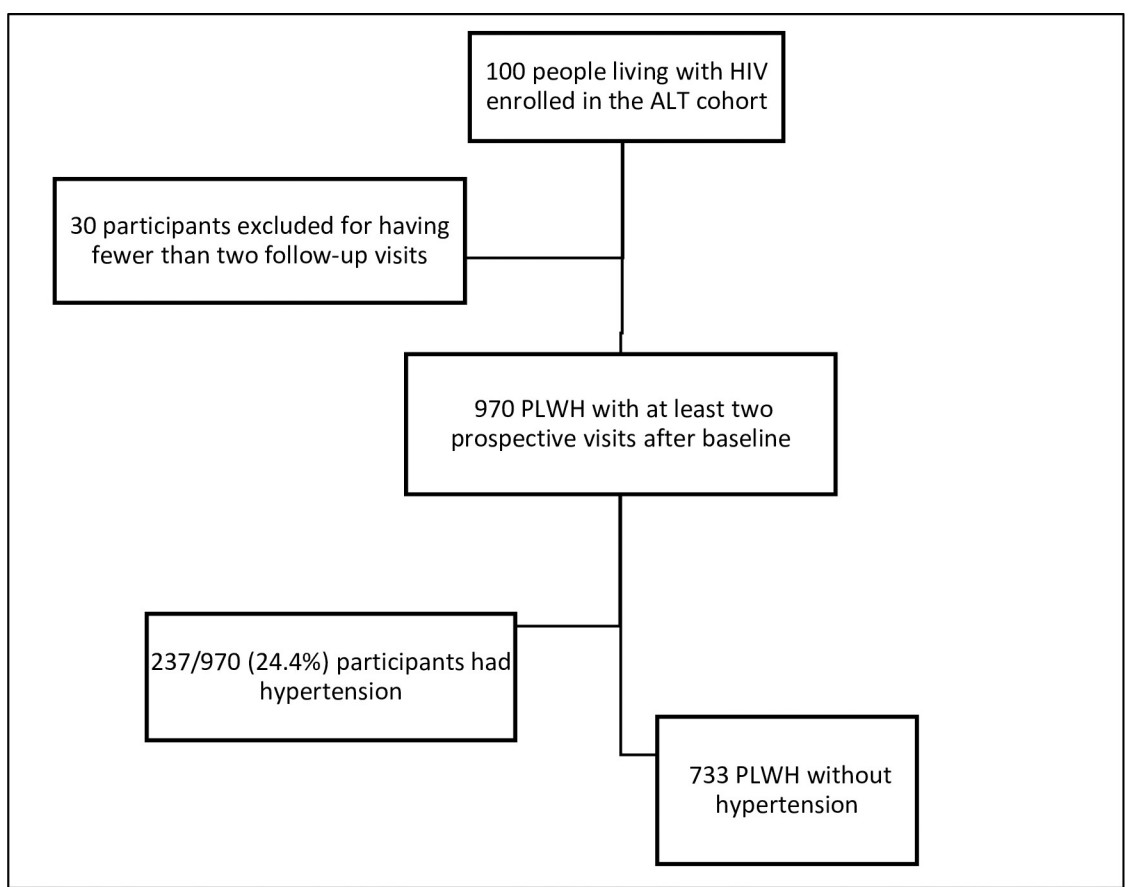

**Fig 1. Participant inclusion for hypertension and weight gain analysis.**

consisted of NNRTIs in 583 (79.5%), and the remaining 150 (20.5%) received PI-based regimens. The NRTI backbones were zidovudine/lamivudine (AZT/3TC) (486, 66.3%), tenofovir disoproxil fumarate (TDF) with emtricitabine or lamivudine (TDF/XTC) (243, 33.2%), and abacavir/lamivudine (ABC/3TC) (4, 0.6%) (Table 3).

A total of 116 participants (15.8%) developed hypertension during 4671.3 person-years of follow-up, with an incidence rate of 24.8 per 1000 person-years (95% CI 20.7–29.8). The risk of developing hypertension increased with increasing follow-up (S1 Fig). Of the 116 participants with incident hypertension, 93 (80.2%) initiated antihypertensive therapy. Participants with obesity were 62% more likely to develop hypertension than those with normal body mass index (BMI) (aIRR 1.80, 95% CI 1.40–2.81), while the risk of hypertension was 1.80 higher in participants with eGFR <90 mL/min/1.73 m$^2$ compared to those with normal renal function (aIRR 1.89, 95% CI 1.20–4.56). The risk of hypertension also increased by 12% (95% CI, 10%–25%) for each 5-year increase in age. There was no evidence to suggest that the association between BMI and hypertension differed according to sex (interaction P = 0.565) or age (interaction P = 0.496). In addition, participants with diastolic blood pressure >80 mmHg and/or systolic blood pressure >120 mmHg were more likely to develop hypertension during follow-up. There was no relationship between exposure to individual antiretroviral agents and the incidence of hypertension (Table 4).

**Table 1. Characteristics of participants with versus without hypertension at cohort enrolment (n = 970).**

| Variable | | Prevalent Hypertension | | No Hypertension* | | All | |
|---|---|---|---|---|---|---|---|
| | | N | % | N | % | n | % |
| | | **237** | **24.4** | **733** | **75.57** | **970** | **100** |
| Sex | Male | 101 | 42.6 | 270 | 36.8 | 371 | 38.3 |
| | Female | 136 | 57.4 | 463 | 63.2 | 599 | 61.8 |
| Prior AIDS | Yes | 113 | 47.7 | 353 | 48.2 | 466 | 48.0 |
| | No | 124 | 52.3 | 380 | 51.8 | 504 | 52.0 |
| Hepatitis B infection | Positive | 8 | 3.4 | 31 | 4.2 | 39 | 4.0 |
| | Negative | 229 | 96.6 | 702 | 95.8 | 931 | 96.0 |
| Smoking Status | Current | 3 | 1.3 | 18 | 2.5 | 21 | 2.2 |
| | Prior | 50 | 21.1 | 149 | 20.3 | 199 | 20.5 |
| | Never | 184 | 77.6 | 566 | 77.2 | 750 | 77.3 |
| Alcohol Status | Current | 52 | 21.9 | 188 | 25.7 | 240 | 24.7 |
| | Prior | 184 | 77.6 | 530 | 72.3 | 714 | 73.6 |
| | Never | 1 | 0.4 | 15 | 2.1 | 16 | 1.7 |
| Diabetes mellitus | Yes | 21 | 8.9 | 14 | 1.9 | 35 | 3.6 |
| | No | 216 | 91.1 | 719 | 98.1 | 935 | 96.4 |
| NRTI backbone | TDF/3TC | 67 | 28.3 | 243 | 33.2 | 310 | 32.0 |
| | AZT/3TC | 164 | 69.2 | 486 | 66.3 | 650 | 67.0 |
| | ABC/3TC | 4 | 1.7 | 4 | 0.6 | 8 | 0.8 |
| | Other | 2 | 0.84 | 0 | 0 | 2 | 0.2 |
| ART class | NNRTIs | 198 | 83.5 | 582 | 79.4 | 780 | 80.4 |
| | PIs | 39 | 16.5 | 151 | 20.6 | 190 | 19.6 |
| **Variable** | | **Median (IQR)** | **n missing (%)** | **Median (IQR)** | **n missing (%)** | **Median (IQR)** | **n missing (%)** |
| Age (years) | | 47.4 (42.4,52.7) | 0 (0) | 44.7 (40.3,50.3) | 0 (0) | 45.4 (40.4,50.5) | 0 (0) |
| Systolic Blood Pressure | | 130 (120,146) | 0 (0) | 120 (110,125) | 0 (0) | 120 (110,130) | 0 (0) |
| Diastolic Blood Pressure | | 80 (70,90) | 0 (0) | 70 (69,80) | 0 (0) | 70 (70,80) | 0 (0) |
| Body mass index (Kg/M$^2$) | | 23.1 (20.3,26.0) | 5 (2.11) | 22.1 (19.7,25.0) | 8 (1.1) | 22.4 (19.8,25.3) | 13 (1.3) |
| eGFR (mL/min/1.73 m2) | | 116 (103,128) | 86 (36.3) | 122.9 (109,130) | 325 (44.3) | 121.4 (107,130) | 411 (42.4) |
| HIV RNA (copies/mL) | | 19 (19,19) | 0 (0) | 19 (19,19) | 0 (0) | 19 (19,19) | 0 (0) |
| Baseline CD4 (cells/µL) | | 518 (378,689) | 0 (0) | 508 (359,683) | 0 (0) | 509 (365,684) | 0 (0) |
| Nadir CD4 (cells/µL) | | 391 (275,507) | 0 (0) | 401 (290,528) | 0 (0) | 399 (289,523) | 0 (0) |
| TCHOL (mmol/L) | | 4.9 (4.3,5.7) | 2 (0.8) | 4.7 (4.0,5.3) | 12 (1.6) | 4.7 (4.1,5.4) | 14 (1.4) |
| LDL (mmol/L) | | 2.8 (2.2,3.3) | 2 (0.8) | 2.6 (2,3.2) | 11 (1.5) | 2.6 (2.0,3.2) | 13 (1.3) |
| HDL (mmol/L) | | 1.2 (1.0,1.5) | 2 (0.8) | 1.2 (1.0,1.5) | 11 (1.5) | 1.2 (0.98,1.5) | 13 (1.3) |
| TRIG (mmol/L) | | 1.4 (1.1,1.9) | 94 (39.7) | 1.3 (0.9,1.8) | 358 (48.8) | 1.3 (1.0,1.8) | 452 (46.6) |
| Number of Follow-up visits | | 5 (4,6) | 0 (0) | 5 (4,6) | 0 (0) | 5 (4,6) | 0 (0) |
| Baseline date (mm/yy) | | 02/15 (08/14,05/15) | 0 (0) | 08/14 (05/15,06/15) | 0 (0) | 03/15 (08/14,06/15) | 0 (0) |
| **Exposure to ART class/NRTIS (years)** | | **Median (IQR)** | **n exposed (%)** | **Median (IQR)** | **n exposed (%)** | **Median (IQR)** | **n exposed (%)** |
| Cumulative exposure to NNRTIs | | 9.4 (7.7,9.6) | 234 (98.7) | 9.3 (4.9,9.6) | 721 (98.4) | 9.3 (5.4,9.6) | 955 (98.5) |
| Cumulative exposure to PIs | | 6.3 (3.0,7.3) | 37 (15.6) | 4.9 (2.2,6.7) | 139 (19.0) | 5.4 (2.4,6.9) | 176 (18.1) |
| Cumulative exposure to ddI | | 5.9 (2.7,7.1) | 11 (4.6) | 6.8 (5.2,7.4) | 28 (3.8) | 6.0 (5.1,7.4) | 39 (4.0) |
| Cumulative exposure to d4T | | 2.7 (2.1,3.2) | 201 (84.8) | 2.6 (2.0,3.1) | 570 (77.8) | 2.6 (2.1,3.1) | 771 (79.5) |
| Cumulative exposure to AZT | | 6.5 (4.9,7.0) | 210 (88.6) | 6.4 (2.4,7.0) | 684 (93.3) | 6.5 (3.0,7.0) | 894 (92.2) |
| Cumulative exposure to TDF | | 5.1 (2.4,6.9) | 64 (27.0) | 3.5 (1.7,6.1) | 230 (31.4) | 3.6 (1.8,6.3) | 294 (30.3) |
| Cumulative exposure to EFV | | 9.3 (6.2,9.6) | 204 (86.1) | 9.2 (4.6,9.6) | 592 (80.8) | 9.3 (4.9,9.6) | 796 (82.1) |
| Cumulative exposure to NVP | | 2.2 (0.1,7.6) | 69 (29.1) | 1.6 (0.1,8.4) | 278 (37.9) | 1.6 (0.1,8.2) | 347 (35.8) |

(*Continued*)

**Table 1.** (Continued)

| Variable | Prevalent Hypertension | | No Hypertension* | | All | |
|---|---|---|---|---|---|---|
| | N | % | N | % | n | % |
| | 237 | 24.4 | 733 | 75.57 | 970 | 100 |
| Cumulative exposure to LPV | 6.4 (5.3,7.1) | 33 (13.9) | 6.0 (4.0,7.2) | 102 (13.9) | 6.1 (4.1,7.2) | 135 (13.9) |
| Cumulative exposure to ATV | 2.1 (1.6,2.4) | 8 (3.4) | 1.4 (0.4,2.1) | 42 (5.7) | 1.6 (0.5,2.2) | 50 (5.2) |

Note: *733 participants without hypertension at baseline were included in the incidence analysis. n means the number of participants; AIDS-prior AIDS-defining event; NNRTIs-non nucleoside reverse transcriptase inhibitors; NRTIs-nucleos(t)ide reverse transcriptase inhibitors, ART-antiretroviral therapy, eGFR-estimated glomerular filtration rate, PI-protease inhibitors, TDF- tenofovir disoproxil fumarate; EFV-efavirenz, LPV-lopinavir; ATV-atazanavir; ddI-didanosine; d4t-stavudine; AZT-zidovudine; NVP-nevirapine; LPV-lopinavir; RTV-ritonavir; ABC-abacavir; 3TC-lamivudine; TRIG-triglycerides; HDL-high-density lipoprotein; LDL-low-density lipoprotein cholesterol; TCHOL-total cholesterol; mm/yy means month followed by the year of the baseline date.

### Data completeness and sensitivity analysis

Overall, there was high data completeness on most variables at baseline except eGFR (411/970, 42.4%) and triglycerides (452/970, 46.6%). The regular Little's MCAR test gives a $\chi^2$ distance of 42.86 with 32 degrees of freedom, P = 0.095. The null hypothesis for Little's MCAR test is that the data are missing completely at random (MCAR). Therefore, we do not reject the null hypothesis, and the test provides evidence that the missing data in the six variables of interest are not MCAR under significance level 0.05. Since data were missing completely at random, there was no justification for using imputing missing data.

## Discussion

This prospective cohort study is among the few to determine the prevalence and incidence of hypertension and its association with exposure to antiretrovirals in PLWH with long ART exposure in SSA. The prevalence of hypertension at enrolment was 24.4% and the factors associated with hypertension were older age, male sex, diabetes mellitus, obesity, and elevated total cholesterol levels. Compared with PLWH without hypertension at enrolment, the odds of prevalent hypertension were also higher in individuals with prior exposure to stavudine and nevirapine. The overall cumulative incidence was 15.8%, with an incidence rate of 24.8 per 1000 person-year of follow-up. Incident hypertension was associated with older age, renal insufficiency, obesity, and blood pressure levels higher than optimal, but not with exposure to antiretroviral drugs.

The incidence of hypertension in the present analysis is higher than that in a previous analysis in the same clinic, which reported a prevalence of 15.1% and an incidence of 19 cases per 1000 person-years eight years before our analysis [36]. However, participants in the present analysis are approximately 10 years older and with longer ART exposure. Increasing rates of hypertension in PLWH have been similarly reported in another cohort of people living with HIV in Uganda [37]. Together, these results suggest increasing rates of hypertension in PLWH in SSA, probably due to ageing and increases in other hypertension risk factors such as ART exposure. While direct comparisons between cohorts cannot be made, the presented incidence of hypertension is within the range of 19 and 54 cases per 1000 person-years, as reported by two studies in Uganda and South Africa, respectively [23, 36]. However, other cohorts in SSA have reported higher incidence rates than those in our analysis [22, 24]. The difference in hypertension rates between cohorts may reflect differences in cohorts as well as differences in hypertension screening. Therefore, HIV programs in SSA should integrate hypertension screening and management into existing ART delivery innovations, as integrated models of

**Table 2. Factors associated with prevalent hypertension in the ALT cohort at baseline (n = 970).**

| Variable | Variable Categories | Crude OR (95% CI) | p-value | Adjusted OR (95% CI) | p-value |
|---|---|---|---|---|---|
| **Sex** | **Female** | **Ref** | | **Ref** | |
| | **Male** | **1.27 (0.95,1.72)** | **0.112** | **1.70 (1.20,2.34)** | **0.002** |
| *Alcohol History* | *Never* | *Ref* | | | |
| | *Current* | *4.15 (0.54,32.14)* | *0.173* | *1.10 (0.19,12.04)* | *0.856* |
| | *Prior* | *5.21 (0.68,39.7)* | *0.111* | *1.67 (0.20,14.42)* | *0.698* |
| **Diabetes mellitus** | **No DM** | **Ref** | | **Ref** | |
| | **DM** | **2.97 (1.48,5.95)** | **0.002** | **2.37 (1.10,4.01)** | **0.011** |
| **Age, Per 5 years increment** | | **1.20 (1.10,1.31)** | **0.000** | **1.18 (1.10,1.34)** | **0.003** |
| **BMI (Kg/M2)** | **<18.5** | **0.98 (0.62,1.54)** | **0.922** | **0.91 (0.56,1.50)** | **0.567** |
| | **18.5–24.9** | **Ref** | | **Ref** | |
| | **25–29.9** | **1.33 (0.92,1.92)** | **0.130** | **1.47 (0.98,2.30)** | **0.075** |
| | **>29.9** | **1.66 (1.15,2.87)** | **0.047** | **1.99 (1.08,3.60)** | **0.031** |
| *GFR (mL/min/1.73 m2)* | *<90* | *1.60 (0.85,2.99)* | *0.142* | *1.13 (0.61,2.46)* | *0.765* |
| | *≥90* | Ref | | *Ref* | |
| | *Missing* | *0.75 (0.55,1.02)* | *0.063* | *1.43 (0.95,2.09)* | *0.078* |
| *HIV RNA (copies/mL)* | *<200* | *1.74 (0.81,3.76)* | *0.159* | *1.78 (0.71,4.00)* | *0.173* |
| | *≥200* | Ref | | Ref | |
| *Baseline TRIG (mmol/L)* | *<1.7* | *Ref* | | *Ref* | |
| | *1.7–2.2* | *1.07 (0.58,1.95)* | *0.837* | *0.83 (0.50,1.60)* | *0.680* |
| | *≥2.3* | *1.41 (0.86,2.30)* | *0.169* | *1.17 (0.69,1.96)* | *0.574* |
| | *Missing* | *0.74 (0.53,1.03)* | *0.071* | *2.94 (2.00,4.69)* | *<0.001* |
| *Baseline LDL (mmol/L)* | *<2.6* | *Ref* | | *Ref* | |
| | *2.6–3.3* | *1.11 (0.79,1.56)* | *0.547* | *0.98 (0.65,1.54)* | *0.861* |
| | *≥3.4* | *1.50 (1.02,2.19)* | *0.037* | *1.10 (0.49,1.99)* | *0.900* |
| | *Missing* | *0.63 (0.14,2.88)* | *0.549* | *0.60 (0.19,4.72)* | *0.660* |
| **Baseline TCHOL (mmol/L)** | **<5.2** | **Ref** | | | |
| | **≥5.2** | **1.57 (1.15,2.12)** | **0.004** | **1.47 (1.16,2.01)** | **0.008** |
| | **Missing** | **0.60 (0.13,2.71)** | **0.505** | **0.48 (0.46,2.80)** | **0.460** |
| **Prior exposure to d4T** | **Never exposed** | **Ref** | | **Ref** | |
| | **Prior Exposure** | **1.60 (1.08,2.37)** | **0.020** | **2.10 (1.35,3.52)** | **<0.001** |
| *Prior exposure to TDF* | *Never exposed* | *Ref* | | *Ref* | |
| | *Prior Exposure* | *0.81 (0.58,1.12)* | *0.200* | *0.76 (0.61,1.34)* | *0.489* |
| *Prior exposure to EFV* | *Never exposed* | *Ref* | | *Ref* | |
| | *Prior Exposure* | *0.73 (0.51,1.07)* | *0.104* | *0.75 (0.51,1.13)* | *0.144* |
| **Prior exposure to NVP** | **Never exposed** | **Ref** | | **Ref** | |
| | **Prior Exposure** | **1.47 (0.98,2.21)** | **0.062** | **1.90 (1.25,3.01)** | **0.002** |
| *Prior exposure to ATV* | *Never exposed* | *Ref* | | *Ref* | |
| | *Prior Exposure* | *0.61 (0.28,1.31)* | *0.203* | *0.86 (0.40,2.14)* | *0.710* |

Note: OR-odds ratio, AIDS-prior AIDS-defining event; BMI-body mass index; eGFR-estimated glomerular filtration rate; TDF-tenofovir disoproxil fumarate; EFV-efavirenz; ATV-atazanavir; d4T-stavudine; NVP-nevirapine, TRIG-triglycerides, HDL-high-density lipoprotein; LDL-low-density lipoprotein cholesterol; TCHOL-total cholesterol. Covariates prior AIDS, hepatitis B infection status, smoking history, prior exposure to lopinavir (LPV), abacavir (ABC), didanosine (ddI), high-density lipoprotein, baseline year, and baseline and nadir CD4 counts were not significant and were not confounders in the univariable analysis (P>0.2) and were not considered in the multivariable analysis. The significant variables in the final multivariable model are in bold, whereas the insignificant variables are in italics.

**Table 3. Characteristics of participants without hypertension at baseline (n = 733).**

| | | Incident Hypertension | | No incident Hypertension | | All | |
|---|---|---|---|---|---|---|---|
| | | **n** | **%** | **n** | **%** | **n** | **%** |
| | | **116** | **15.8** | **617** | **84.2** | **733** | **100** |
| Sex | Male | 43 | 37.1 | 227 | 36.8 | 270 | 36.8 |
| | Female | 73 | 62.9 | 390 | 63.2 | 463 | 63.2 |
| Prior AIDS | Yes | 46 | 39.7 | 307 | 49.8 | 353 | 48.2 |
| | No | 70 | 60.3 | 310 | 50.2 | 380 | 51.8 |
| Hepatitis B infection | Positive | 4 | 3.5 | 27 | 4.4 | 31 | 4.2 |
| | Negative | 112 | 96.6 | 590 | 95.6 | 702 | 95.8 |
| Smoking Status | Current | 3 | 2.6 | 15 | 2.4 | 18 | 2.5 |
| | Prior | 16 | 13.8 | 133 | 21.6 | 149 | 20.3 |
| | Never | 97 | 83.6 | 469 | 76.0 | 566 | 77.2 |
| Alcohol Status | Current | 24 | 20.7 | 164 | 26.6 | 188 | 25.7 |
| | Prior | 89 | 76.7 | 441 | 71.5 | 530 | 72.3 |
| | Never | 3 | 2.6 | 12 | 1.9 | 15 | 2.1 |
| Diabetes mellitus | Yes | 3 | 2.6 | 11 | 1.8 | 14 | 1.9 |
| | No | 69 | 59.5 | 358 | 58.0 | 427 | 58.3 |
| | Missing | 44 | 37.9 | 248 | 40.2 | 292 | 39.8 |
| NRTI backbone | TDF/3TC | 30 | 25.9 | 213 | 34.5 | 243 | 33.2 |
| | AZT/3TC | 86 | 74.14 | 400 | 64.8 | 486 | 66.3 |
| | ABC/3TC | 0 | 0 | 4 | 0.7 | 4 | 0.6 |
| ART regimen | NNRTIs | 103 | 88.8 | 480 | 77.8 | 583 | 79.5 |
| | PIs | 13 | 11.2 | 137 | 22.2 | 150 | 20.5 |
| | | **Median (IQR)** | **n missing (%)** | **Median (IQR)** | **n missing (%)** | **Median (IQR)** | **n missing (%)** |
| Age (years) | | 45.6 (42.2,51.7) | 0 (0) | 44.4 (40,49.7) | 0 (0) | 44.7 (40.3,50.3) | 0 (0) |
| Systolic Blood Pressure | | 130 (120,140) | 0 (0) | 112 (110,120) | 0 (0) | 120 (110,125) | 0 (0) |
| Diastolic Blood Pressure | | 80 (70,85.5) | 0 (0) | 70 (67,74) | 0 (0) | 70 (69,80) | 0 (0) |
| BMI (Kg/M$^2$) | | 23.2 (21.1,27.1) | 0 (0) | 21.8 (19.5,24.9) | 8 (1.30) | 22.1 (19.7,25.0) | 8 (1.1) |
| GFR (mL/min/1.73 m2) | | 121 (103,126) | 43 (37.1) | 123 (110,130) | 282 (45.7) | 122.9 (109,130) | 325 (44.3) |
| HIV RNA (copies/mL) | | 19 (19,19) | 0 (0) | 19 (19,19) | 0 (0) | 19 (19,19) | 0 (0) |
| Baseline CD4 (cells/µL) | | 534 (373.5,679) | 0 (0) | 506 (350,687) | 0 (0) | 508 (359,683) | 0 (0) |
| Nadir CD4 (cells/µL) | | 411 (290,581.5) | 0 (0) | 399 (290,519) | 0 (0) | 401 (290,528) | 0 (0) |
| TCHOL (mmol/L) | | 4.7 (4.2,5.4) | 3 (2.6) | 4.6 (4.0,5.3) | 9 (1.5) | 4.7 (4.0,5.3) | 12 (1.6) |
| LDL (mmol/L) | | 2.7 (2.1,3.2) | 3 (2.6) | 2.6 (2.0,3.1) | 8 (1.3) | 2.6 (2.0,3.2) | 11 (1.5) |
| HDL (mmol/L) | | 1.3 (1.0,1.5) | 3 (2.6) | 1.2 (1.0,1.5) | 8 (1.3) | 1.2 (1.0,1.5) | 11 (1.5) |
| TRIG (mmol/L) | | 1.5 (1.0,2.2) | 49 (42.2) | 1.3 (0.9,1.7) | 309 (50.1) | 1.3 (0.9,1.8) | 358 (48.8) |
| Number of Follow-up visits | | 5 (5,6) | 0 (0) | 5 (4,6) | 0 (0) | 5 (4,6) | 0 (0) |
| Baseline date (mm/yy) | | 02/15 (08/14,06/15) | 0 (0) | 03/15 (08/14,06/15) | 0 (0) | 08/14 (05/15,06/15) | 0 (0) |
| **Exposure to ART class (years)** | | **Median (IQR)** | **n exposed (%)** | **Median (IQR)** | **n exposed (%)** | **Median (IQR)** | **n exposed (%)** |
| Cumulative exposure to NNRTIs | | 9.4 (9.1,9.6) | 115 (99.1) | 9.3 (3.9,9.6) | 606 (98.2) | 9.3 (4.9,9.6) | 721 (98.4) |
| Cumulative exposure to PIs | | 7.9 (3.0,8.1) | 11 (9.5) | 4.6 (2.2,6.5) | 128 (20.8) | 4.9 (2.2,6.7) | 139 (19.0) |
| Cumulative exposure to ddI | | 7.1 (6.0,7.6) | 6 (5.2) | 6.0 (5.2,7.4) | 22 (3.6) | 6.2 (5.2,7.4) | 28 (3.8) |
| Cumulative exposure to d4T | | 2.6 (2.1,3.1) | 93 (80.2) | 2.6 (2.0,3.0) | 477 (77.3) | 2.6 (2.0,3.1) | 570 (77.8) |
| Cumulative exposure to AZT | | 6.7 (4.1,7.0) | 109 (94.0) | 6.3 (2.0,7.0) | 575 (93.2) | 6.4 (2.4,7.0) | 684 (93.3) |
| Cumulative exposure to TDF | | 3.1 (1.1,6.9) | 25 (21.6) | 3.5 (1.8,6.0) | 205 (33.2) | 3.5 (1.7,6.1) | 230 (31.4) |
| Cumulative exposure to EFV | | 9.4 (9.1,9.6) | 97 (83.6) | 9.2 (3.8,9.5) | 495 (80.2) | 9.2 (4.6,9.6) | 592 (80.8) |
| Cumulative exposure to NVP | | 0.7 (0.1,7.6) | 33 (28.5) | 1.6 (0.1,8.4) | 245 (39.7) | 1.6 (0.1,8.4) | 278 (37.9) |
| Cumulative exposure to LPV | | 8.1 (7.8,8.4) | 9 (7.8) | 5.8 (3.9,6.8) | 93 (15.1) | 6.0 (4.0,7.2) | 102 (13.9) |

*(Continued)*

**Table 3.** (Continued)

| | Incident Hypertension | | No incident Hypertension | | All | |
|---|---|---|---|---|---|---|
| | n | % | n | % | n | % |
| | **116** | **15.8** | **617** | **84.2** | **733** | **100** |
| Cumulative exposure to RTV | 7.9 (3.0,8.4) | 11 (9.5) | 4.6 (2.2,6.5) | 128 (20.8) | 4.9 (2.2,6.7) | 139 (19.0) |
| Cumulative exposure to ATV | 0.8 (0.4,1.3) | 2 (1.7) | 1.5 (0.4,2.2) | 40 (6.5) | 1.4 (0.4,2.1) | 42 (5.7) |

Note: AIDS means prior AIDS-defining event, NRTI-nucleos(t)ide reverse transcriptase inhibitors, ART-antiretroviral therapy, BMI-body mass index, eGFR-estimated glomerular filtration rate, NNRTIs-non-nucleoside reverse transcriptase inhibitors, PI-protease inhibitors, TDF-tenofovir disoproxil fumarate, EFV-efavirenz, LPV-lopinavir, ATV-atazanavir, ddI-didanosine, d4T-stavudine, AZT-zidovudine, NVP-nevirapine, LPV-lopinavir, RTV-ritonavir, ABC-abacavir, 3TC-lamivudine, TRIG-triglycerides, HDL-high-density lipoprotein, LDL-low-density lipoprotein cholesterol, TCHOL-total cholesterol, mm/yy-month/year.

hypertension/HIV care have been demonstrated to be feasible even in public facilities in SSA [38, 39].

Our findings are also consistent with analyses of other cohorts in SSA that reported male sex, old age, low nadir CD4 count, and obesity as risk factors for hypertension [22]. Therefore, PLWH with these characteristics should be closely monitored. Previous exposure to nevirapine and stavudine has also been associated with risk or progression in several studies [13, 22, 40, 41]. Although contemporary ART regimens are increasingly available in most SSA countries, participants with prior exposure to these toxic agents should be targeted for hypertension and cardiovascular disease screening. The exact mechanism underlying hypertension in people with prior exposure to nevirapine or thymidine analogues is unclear but may be related to metabolic changes. In the AGEhIV cohort, the association between stavudine exposure and hypertension was attenuated after adjustment for lipodystrophy [13], suggesting that lipodystrophy may have an underlying role. The mechanism underlying the elevated risk of hypertension associated with nevirapine use remains unclear and should be investigated further.

Our study analysis several limitations. First, there is potential residual confounding because we could not measure and adjust for other hypertension risk factors, such as family history, medication use, unhealthy diet, and physical inactivity. In addition, data on laboratory parameters were only available at baseline and could not be modelled as time-updated covariates. Second, we cannot rule out unmeasured biases, as the analysis is based on an observational cohort. Furthermore, blood pressure was only measured annually, and this may lead to, and this may lead to inaccurate estimation of blood pressure incident dates. However, most hypertension guidelines recommend at least one annual blood pressure measurement per year. Third, we did not determine the incidence of cardiovascular disease, an important complication of uncontrolled hypertension. However, the detection and management of cardiovascular disease remains a challenge for many HIV programs in SSA countries [8, 42]. Fourth, missing data for some variables could have resulted in bias, although the proportion of participants with missing data was consistently small across the variables, and data was completely missing at random. Finally, in our analysis, we could not determine the hypertension risk posed by contemporary ART regimens, such as INSTIs, as these agents were recently introduced in Uganda; therefore, such an analysis would be underpowered. Future studies should investigate the risk of hypertension posed by contemporary regimens in SSA cohorts since there are reports that agents are differentially associated with a greater risk of weight gain in black people [43].

Despite these limitations, our results may inform the improvement of hypertension and HIV-care programs in SSA. With the increasing survival of PLWH and the increasing burden of cardiovascular disease, screening and managing cardiovascular risk factors, such as

**Table 4. Factors associated with of incident hypertension in the ALT cohort (n = 733).**

| Variable | Variable categories | (Events/PYFU) | Incidence Rate/1000 PY (95% CI) | Crude IRR (95% CI) | p-value | Adjusted IRR (95% CI) | p-value |
|---|---|---|---|---|---|---|---|
| Prior AIDS | Yes | 46/2263 | 20.3 (15.2,27.1) | 0.70 (0.48,1.01) | 0.060 | 0.93 (0.58,1.20) | 0.579 |
| | No | 70/2408.3 | 29.1 (23.0,36.7) | Ref | | Ref | |
| NRTI | TDF/3TC | 30/1538.5 | 19.5 (13.6,27.9) | Ref | | Ref | |
| | AZT/3TC | 86/3103.6 | 27.7 (22.4,34.2) | 1.42 (0.94,2.15) | 0.098 | 1.24 (0.83,1.90) | 0.340 |
| | ABC/3TC | 0/29.2 | 0 | - | | - | |
| ART Class | NNRTIs | 103/3705.6 | 27.8 (22.7,33.4) | 2.04 (1.14,3.63) | 0.016 | 1.50(0.90,2.70) | 0.150 |
| | **PIs** | **13/962.6** | **13.5 (7.8,23.3)** | **Ref** | | **Ref** | |
| **Age** | **Per 5 years** | **116/4671.3** | **24.8 (20.7,29.8)** | **1.19 (1.07,1.32)** | **0.002** | **1.12 (1.10,1.25)** | **0.009** |
| **BMI (Kg/m2)** | **<18.5** | **12/638.5** | **18.8 (10.7,33.1)** | **0.82 (0.44,1.51)** | **0.516** | **0.89(0.50,1.64)** | **0.670** |
| | **18.5–24.9** | **66/2864.3** | **23.0 (18.1,29.3)** | **Ref** | | **Ref** | |
| | **25–29.9** | **22/888.4** | **24.8 (16.3,37.6)** | **1.07 (0.66,1.74)** | **0.770** | **0.87 (0.53,1.41)** | **0.567** |
| | **>29.9** | **16/280** | **57.1 (35.0,93.3)** | **2.48 (1.44,4.28)** | **0.001** | **1.80 (1.40,2.81)** | **0.009** |
| **GFR (mL/min/1.73 m2)** | **<90** | **8/193.1** | **41.4 (20.7,82.8)** | **1.56 (0.75,3.25)** | **0.236** | **1.89 (1.20,4.56)** | **0.017** |
| | **≥90** | **65/2446.8** | **26.6 (20.8,33.9)** | **Ref** | | **Ref** | |
| | **Missing** | **43/2031.4** | **21.2 (15.7,28.5)** | **0.80 (0.54,1.17)** | **0.248** | **0.70 (0.49,1.10)** | **0.112** |
| Nadir CD4 (cells/μL) | <200 | 9/411.6 | 21.9 (11.4,42.0) | 0.69 (0.34,1.41) | 0.310 | 0.66 (0.36,1.38) | 0.257 |
| | 200–349 | 33/1379 | 23.9 (17.0,33.7) | 0.76 (0.48,1.18) | 0.221 | 0.74 (0.48,1.17) | 0.170 |
| | 350–500 | 29/1460.2 | 19.9 (13.8,28.6) | 0.63 (0.39,1.00) | 0.050 | 0.80(0.51,1.26) | 0.250 |
| | >500 | 45/1420.5 | 31.7 (23.7,42.4) | Ref | | Ref | |
| Baseline Year | 2014 | 52/1829.8 | 28.4 (21.7,37.3) | 1.26 (0.88,1.82) | 0.213 | 1.19 (0.80,1.77) | 0.277 |
| | 2015 | 64/2841.5 | 22.5 (17.6,28.8) | Ref | | Ref | |
| **Systolic blood pressure (mmHg)** | **<120** | **17/2399.5** | **7.1 (4.4,11.4)** | **Ref** | | **Ref** | |
| | **120–129** | **27/1289.1** | **20.9 (14.4,30.5)** | **2.96 (1.61,5.42)** | **<0.001** | **2.34 (1.31,4.33)** | **0.006** |
| | **≥130** | **72/982.7** | **73.3 (58.2,92.3)** | **10.34 (6.1,17.54)** | **<0.001** | **5.43(3.20,9.41)** | **<0.001** |
| **Diastolic blood pressure (mmHg)** | **<80** | **43/3512.4** | **12.2 (9.1,16.5)** | **Ref** | | **Ref** | |
| | **80–84** | **42/705.6** | **59.5 (44,80.5)** | **4.86 (3.18,7.44)** | **<0.001** | **2.83 (1.82,4.44)** | **<0.001** |
| | **≥85** | **31/453.3** | **68.4 (48.1,97.2)** | **5.59 (3.52,8.86)** | **<0.001** | **2.39 (1.43,4.10)** | **<0.001** |
| Prior exposure to PIs | Never exposed | 105/3787.7 | 27.7 (22.9,33.6) | Ref | | Ref | |
| | Prior Exposure | 11/883.6 | 12.4 (6.9,22.5) | 0.45 (0.24,0.84) | 0.012 | 0.79 (0.43,1.56) | 0.256 |
| Prior exposure to TDF | Never exposed | 91/3227.9 | 28.2 (23.0,34.6) | Ref | | Ref | |
| | Prior Exposure | 25/1443.4 | 17.3 (11.7,25.6) | 0.61 (0.39,0.96) | 0.031 | 0.69 (0.56,1.38) | 0.200 |
| Prior exposure to EFV | Never exposed | 98/3596.8 | 27.2 (22.4,33.2) | Ref | | Ref | |
| | Prior Exposure | 18/1074.5 | 16.8 (10.6,26.6) | 0.61 (0.37,1.02) | 0.058 | 0.98 (0.57,1.58) | 0.771 |
| Prior exposure to LPV | Never exposed | 107/4020 | 26.6 (22.0,32.2) | Ref | | Ref | |
| | Prior Exposure | 9/651.3 | 13.8 (7.2,26.6) | 0.52 (0.26,1.03) | 0.059 | 0.59(0.34,1.36) | 0.276 |
| Prior exposure to ATV | Never exposed | 114/4415.8 | 25.8 (21.5,31.0) | Ref | | Ref | |
| | Prior Exposure | 2/255.5 | 7.8 (2.0,31.3) | 0.30 (0.07,1.23) | 0.094 | 1.00(0.70, 1.41) | 0.936 |

Note: * PYFU-person-years of follow-up; AIDS-prior AIDS-defining event; NRTI-nucleos(t)ide reverse transcriptase inhibitors; ART-antiretroviral therapy; BMI-body mass index; GFR-glomerular filtration rate; PI-protease inhibitors; TDF-tenofovir disoproxil fumarate; EFV-efavirenz; LPV-lopinavir; ATV-atazanavir; ABC-abacavir; 3TC-lamivudine. "Prior exposure to PIs" was collinear with "Prior exposure to LPV" (VIF = 6.9). However, these variables were insignificant in the final model (even when included one at a time), and they were dropped. The adjusted estimates for these variables were obtained by adding the variables back into the adjusted model, one at a time. Covariates sex, prior AIDS, hepatitis B infection status, smoking history, alcohol history, diabetes mellitus, prior exposure to didanosine (ddI), stavudine (d4t), zidovudine (AZT), nevirapine (NVP), non-nucleoside reverse transcriptase inhibitors (NNRTIs), baseline lipid values (triglycerides, high-density lipoprotein, low-density lipoprotein cholesterol and total cholesterol), HIV RNA, and baseline CD4 counts were all non-significant and were not confounders in the univariable analysis ($P>0.2$) and were not considered in the multivariable analysis. The significant covariates are indicated in bold.

hypertension, will become essential for preventing cardiovascular deaths, especially in SSA [6]. In the analysed cohort, the rates of other cardiovascular risk factors, such as diabetes and smoking, were low, signifying that hypertension may play an even more critical role as a cardiovascular risk factor in SSA. The low smoking rates are consistent with reports from other SSA HIV cohorts in which smoking rates are generally low in PLWH, especially in women [44]. In addition, data from longitudinal cohorts on the relationship between hypertension and antiretroviral exposure in PLWH with long ART exposure in SSA are lacking. Our cohort enrolled 1000 participants who had been on ART for at least 10 years at baseline and were subsequently followed up for almost eight years. Therefore, this analysis contributes to the evidence for screening for hypertension and its risk factors in PLWH with prolonged exposure to ART in SSA. Importantly, highlight the importance of blood pressure screening, supports lipid and glucose testing, and screening for renal insufficiency in heavily treated PLWH. Furthermore, this analysis has several strengths. This cohort had high retention, as 97% of the participants had at least two follow-up visits, thus minimising attrition bias. In addition, we used data from people with prolonged ART exposure, who reflect the future HIV population. Finally, the analysis was based on routinely collected clinical data and may represent real-life urban settings in Uganda.

In conclusion, the prevalence and incidence of hypertension in this cohort of heavily treated PLWH was higher than those previously reported in the same HIV clinic. In addition, hypertension was associated with traditional risk factors and exposure to older toxic antiretroviral agents. Collectively, our results highlight the need to monitor hypertension in PLWH with prolonged exposure to ART and the need for HIV programs in SSA to integrate hypertension screening into existing HIV treatment programs.

## Supporting information

**S1 Fig. Incidence rate of hypertension with increasing follow-up time.** The rates of hypertension increased with increasing follow-up duration.
(TIF)

## Acknowledgments

The authors declare that they have no conflicts of interest. We thank all study participants for contributing data and all staff for coordinating the study. DBM conceived the analysis, developed the analysis plan, conducted the analysis, and wrote the first draft of the manuscript under the supervision of BC, KP, ML, and MP. BC, JM, and RPR are investigators for the ALT cohort and supervised data curation. All the authors have read and approved the final manuscript.

## Author Contributions

**Conceptualization:** Dathan M. Byonanebye.

**Data curation:** Dathan M. Byonanebye.

**Formal analysis:** Dathan M. Byonanebye.

**Funding acquisition:** Rosalind Parkes-Ratanshi, Barbara Castelnuovo.

**Investigation:** Barbara Castelnuovo.

**Project administration:** Joseph Musaazi.

**Supervision:** Kathy Petoumenos, Barbara Castelnuovo.

**Writing – original draft:** Dathan M. Byonanebye.

**Writing – review & editing:** Mark N. Polizzotto, Rosalind Parkes-Ratanshi, Joseph Musaazi, Kathy Petoumenos, Barbara Castelnuovo.

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
