## [Decision Letter · Decision Letter 0]

24 Nov 2022

PONE-D-22-29381Prevalence and incidence of hypertension in a heavily treatment-experienced cohort of people living with HIVPLOS ONE

Dear Dr. Byonanebye,

Thank you for submitting your manuscript to PLOS ONE. After careful consideration, we feel that it has merit but does not fully meet PLOS ONE’s publication criteria as it currently stands. Therefore, we invite you to submit a revised version of the manuscript that addresses the points raised during the review process.

The paper that you submitted to the PLOS ONE has been seen by three referees whose reports are listed below. As you will see, the referees raised several important criticisms which make the paper unacceptable for publication in its present form. This manuscript needs substantial improvement in terms of methodology, writing and sending the message. However, if you can deal with referees' comments and modify the paper according to their suggestions the Editorial Board may reconsider your work. Please submit your revised manuscript by January 08,2023. If you will need more time than this to complete your revisions, please reply to this message or contact the journal office at plosone@plos.org. Please include the following items when submitting your revised manuscript:A rebuttal letter that responds to each point raised by the academic editor and reviewer(s). You should upload this letter as a separate file labeled 'Response to Reviewers'.A marked-up copy of your manuscript that highlights changes made to the original version. You should upload this as a separate file labeled 'Revised Manuscript with Track Changes'.An unmarked version of your revised paper without tracked changes. You should upload this as a separate file labeled 'Manuscript'.

We look forward to receiving your revised manuscript.

Kind regards,

Giuseppe Vittorio De Socio, MD, PhD

Academic Editor

PLOS ONE

Reviewers' comments:

Reviewer's Responses to Questions

**Comments to the Author**

1. Is the manuscript technically sound, and do the data support the conclusions?

Reviewer #1: Partly

Reviewer #2: Partly

Reviewer #3: No

2. Has the statistical analysis been performed appropriately and rigorously? 

Reviewer #1: No

Reviewer #2: No

Reviewer #3: No

3. Have the authors made all data underlying the findings in their manuscript fully available?

Reviewer #1: No

Reviewer #2: No

Reviewer #3: No

4. Is the manuscript presented in an intelligible fashion and written in standard English?

Reviewer #1: Yes

Reviewer #2: Yes

Reviewer #3: Yes

5. Review Comments to the Author

Reviewer #1: General comments:

This manuscript endeavors to estimating the prevalence and incidence of hypertension in a cohort of PLWH from an HIV clinic in Kampala, Uganda.

The methods for this manuscript have a few holes or confusing pieces and I've tried to make comments below to improve the manuscript.

My biggest general comment is that I don't know why both prevalence and incidence are included. More specifically, I question the model looking at incidence. The methods suggest that the only difference between the two models is that people without hypertension at baseline were included from the incidence model. I wonder how well that baseline measurement characterizes those with hypertension. Was any past history information on hypertension included into the assessment? What are the random fluctuations and measurement error for SBP and DBP? These could impact the classification of baseline hypertension. If there are misclassifications, this could lead to people being included in the analyses that already have hypertension and lead to an overestimation of hypertension in the follow-up.

In addition, since the cohort does not appear to be systematically selected, I'm not entirely sure about some of the conclusions. For instance, age has an association with hypertension but I don't know if that's true or an artifact of the selection.

Specific comments:

1. (lines 123-132) I am sure I follow how you performed your variable selection here and have three comments about this section. First, stepwise variable selection procedures, which include backward selection, usually do not do a good job of finding the most appropriate model (e.g., https://doi.org/10.1002/sim.3943). Stepwise procedures and any p-value based selection have quite a bit of evidence suggesting that they are poor at selecting the appropriate variables. For a decent summary, see the link above. Second, Sun et al. (http://dx.doi.org/10.1016/0895-4356(96)00025-X) found that bivariate screening can miss a variable that may be a confounder even when a p-value is as high as what you have (0.25). Bivariate screening can be considered a form of stepwise variable selection, which usually do not do a good job of finding the most appropriate model (e.g., https://doi.org/10.1002/sim.3943). Third, I don't understand how AIC is involved the selection. You've already used stepwise procedures to select variables, so what are you comparing? Generally it's better to perform the entire selection based on more robust criteria, especially measures which assess the fit of the model (such as AIC or BIC where you could you Burnham and Anderson (doi: 10.1177/0049124104268644) as a guide) or, better yet, a shrinkage-based estimator such as lasso or lars.

2. (lines 133-137) First off, it is possible to test to see if missing data are missing completely at random (https://doi.org/10.2307/2290157). I would do that first and, if the data are MCAR, not worry about imputation. If the MCAR test fails, then I would promote using the MICE output above the complete case analyses. Finally, for the imputation model to work best, one really needs to include as much data as possible, i.e., include auxiliary data. If you have more data, I encourage you to include that here.

3. Is follow-up time included as an offset in the Poisson model? This is never explicitly stated in the methods so I wanted to check to make sure this was done. If this was not done, this must be done; otherwise you are not calculating IRRs.

4. (Table 2) Editorial note: it's hard to read your table with the variable names centered and no lines or breaks between the variable categories.

5. (Tables 2 and 4) Is the "global p" coming from a Wald type 3 test? I strongly encourage you to include more information on the methodology in the methods section or drop this altogether. If these are Wald type 3 tests, I will admit that I have always found the results from these confusing and I think your analyses here highlight some of those. For instance in Table 2, for most binary variables in the adjusted model, the p-value of the non-reference group and global p are identical but not for Sex. For alcohol history, the two p-values are pretty high (0.945 and 0.702), but the global p is far smaller (0.182). Though, in a roughly similar situation for Baseline LDL, the global p is larger than all the individual comparisons. From my experience, most people will pay attention to the individual categories and the global tests are going to create confusion.

6. (Tables 2,4) Instead of categorizing the continuous variables in the models (e.g., age, BMI, GFR), could you analyze them as continuous, maybe with a spline or other transformation? If you are really interested in characterizing the associations, that would provide more information than categories or assuming linearity (as was done with age in 5-year increments). The categorizations of these variables are not explained in the methods and, if the categorizations don't have clinical or policy relevance, they may lack utility.

Reviewer #2: This study sought to determine the prevalence and incidence of hypertension and its relationship with exposure to specific antiretroviral therapy in a Ugandan cohort of PLWH with long durations of ART.

Minor

1. Define what is meant by “heavily treatment experience”.

2. Multivariable not multivariate.

3. The authors use rate ratios and risk ratios interchangeability yet these differ (Line 127).

4. The interpretation of odds ratios as percentages is not correct. (e.g., odds of prevalent hypertension increase by 16%).

5. Define contemporary ART regimens. Line 76.

6. What was considered as prior exposure to antiretroviral therapy in the staged model building approach line 115.

7. Provide references for definitions and analysis endpoints for example diabetes mellitus, hepatitis B, and chronic kidney disease.

Major

1. Clarify why those with more than two follow-up visits were excluded for the analysis of prevalence.

2. Was adherence to antiretroviral therapy considered in the analysis? How would it affect the estimates?

3. How was Cumulative exposure to first line and second-line ART modeled

4. For the logistic regression model how were the covariates chosen?

5. It seems counterintuitive to remove untreated viral drugs that were not independently associated with hypertension (line 117 -119) yet the objective of this study is to find the association between antiretroviral drugs and hypertension.

6. What was the rationale for choosing the non-ART confounders listed in line 121 - 122?

7. Stepwise variable selection modeling is problematic in many ways including: it yields confidence intervals for effects and predicted values that are falsely narrow (Altman and Andersen, 1989) and biased regression coefficients that need shrinkage (see Tibshirani, 1996).

8. Line 135. What was the evidence for the assumption of MAR?

9. Tamper the discussion and conclusions given that the blood pressure measurements were annual with a high possibility of measurement error and visit-visit variability.

10. Rate ratios on their own cannot be compared across populations. Instead use standardized rates for comparison of incidence rates the current study with those in other populations.

Reviewer #3: 1- The authors wrote that “all variables in this analysis were fixed at baseline”. This rules out the possibility to evaluate time-varying phenomena, such as changes in eGFR. Please, comment on this.

2- The authors raised the issue of missing variables. The proportion of missing data is reported only for some, but not for all, clinical variables, e.g. alcohol consumption. From the analysis of Table 1, there is a huge variance for HR of some variables (e.g. alcohol consumption). Please, add the number of missing data for all the variables.

3- The authors wrote: “Multivariate logistic regression was used to determine the factors associated with hypertension at baseline”. Further information is needed regarding the list of variables included in the multivariate model.

In the legend for Table 2 the authors wrote: “Covariates sex, prior AIDS, hepatitis B infection status, smoking history, prior exposure to lopinavir (LPV), abacavir (ABC), efavirenz (EFV), didanosine (ddI), high-density lipoprotein, baseline year, and baseline and nadir CD4 counts were not significant in the univariate analysis (P>0.2) and were not considered in the multivariate analysis”.

However, (1) sex is significantly associated to HTN prevalence; this should be clarified. (2) Prior exposure to EFV was reported in the table. Please, better specify the criteria adopted to design Table 2.

The authors wrote that multicollinearity was checked and a VIF >5 was used as cut-off. Considering that the whole number of variables examinated in the univariate and subsequent mutlivariate model is 26, how many varables were excluded for multicollinearity?

6. PLOS authors have the option to publish the peer review history of their article (what does this mean?). If published, this will include your full peer review and any attached files.

Reviewer #1: No

Reviewer #2: No

Reviewer #3: No

---

## [Author Response · Author response to Decision Letter 0]

14 Dec 2022

REVIEWER # 1

General comments:

1. General comment 1: This manuscript endeavors to estimating the prevalence and incidence of hypertension in a cohort of PLWH from an HIV clinic in Kampala, Uganda. The methods for this manuscript have a few holes or confusing pieces and I've tried to make comments below to improve the manuscript. My biggest general comment is that I don't know why both prevalence and incidence are included. More specifically, I question the model looking at incidence. The methods suggest that the only difference between the two models is that people without hypertension at baseline were included from the incidence model. I wonder how well that baseline measurement characterises those with hypertension. Was any past history information on hypertension included into the assessment? What are the random fluctuations and measurement error for SBP and DBP? These could impact the classification of baseline hypertension. If there are misclassifications, this could lead to people being included in the analyses that already have hypertension and lead to an overestimation of hypertension in the follow-up.

Response: We thank the reviewer for these comments, and we are happy to provide detailed responses below. 

First, with respect to why both prevalence and incidence are reported. We would like to clarify that is common for studies to determine both the prevalence and incidence of hypertension in cohorts and we provide evidence for this approach in the publications below. There are many publications and reports that report both prevalence and incidence and some of these publications have been in prestigious journals, including PLOS ONE. 

• Brennan AT, Jamieson L, Crowther NJ, Fox MP, George JA, Berry KM, Stokes A, Maskew M, Sanne I, Long L, Cassim N, Rosen S. Prevalence, incidence, predictors, treatment, and control of hypertension among HIV-positive adults on antiretroviral treatment in public sector treatment programs in South Africa. PLoS One. 2018 Oct 3;13(10):e0204020. doi: 10.1371/journal.pone.0204020. PMID: 30281618; PMCID: PMC6169897.

• Lacruz ME, Kluttig A, Hartwig S, Löer M, Tiller D, Greiser KH, Werdan K, Haerting J. Prevalence and Incidence of Hypertension in the General Adult Population: Results of the CARLA-Cohort Study. Medicine (Baltimore). 2015 Jun;94(22):e952. doi: 10.1097/MD.0000000000000952. PMID: 26039136; PMCID: PMC4616348.

• Tu K, Chen Z, Lipscombe LL; Canadian Hypertension Education Program Outcomes Research Taskforce. Prevalence and incidence of hypertension from 1995 to 2005: a population-based study. CMAJ. 2008 May 20;178(11):1429-35. doi: 10.1503/cmaj.071283. PMID: 18490638; PMCID: PMC2374870.

• Sinnott SJ, Smeeth L, Williamson E, Douglas IJ. Trends for prevalence and incidence of resistant hypertension: population based cohort study in the UK 1995-2015. BMJ. 2017 Sep 22;358:j3984. doi: 10.1136/bmj.j3984. PMID: 28939590; PMCID: PMC5609092.

In addition, reporting prevalence and incidence is complimentary and epidemiologically important since it describes the current burden of disease (prevalence) and determines whether the burden of disease is increasing or not (incidence). This approach is important in several ways. First, the HIV population in Sub-Saharan Africa is dynamic and ageing; these data have also not been described before in this setting. For example, antiretroviral therapy is changing, and coverage is increasing (and this is stated in the introduction section (line 61). Therefore, in a dynamic population, determining prevalence alone would not provide enough data to characterise the burden of hypertension and inform programming and policy. Therefore, we hope the reviewer will appreciate the rationale for determining and reporting both the prevalence and incidence of hypertension in our study. 

The second issue raised by the reviewer is "how well that baseline measurement characterises those with hypertension. Was any past history information on hypertension included into the assessment? What are the random fluctuations and measurement error for SBP and DBP?". We would like to clarify that data on prior hypertension diagnosis (and treatment), and blood pressure were available, and this was the basis for defining hypertension prevalence at baseline. Prevalent hypertension, which was determined at baseline, was defined as two consecutive systolic blood pressure (SBP) measures greater than 140 mmHg and/or diastolic blood pressure (DBP) greater than 90 mmHg and/or documented diagnosis and/or the initiation of antihypertensives within one year prior and up to one month after the baseline date. This clarification has been included in the methods section (lines 111-114). 

Regarding random fluctuations and measurement errors for SBP and DBP, our definitions for both prevalent and incident hypertension are based on two consecutive abnormal blood pressure measurements (≥140/90 mmHg) or evidence of antihypertensive treatment. This approach minimises measurement bias and misclassification than if one reading was used. We agree that one BP measure would falsely classify patients with subtle increments in blood pressure due to non-hypertensive events such as anxiety syndromes as having hypertension. For this reason, the definition of hypertension is based on two consecutive readings. The definition is also consistent with international Hypertension diagnosis and management guidelines that recommend two consecutive abnormal blood pressure measurements [1].

Further, hypertension treatment and blood pressure measurement data are standardised according to the clinic standard operating procedures. Blood pressure measurement is done by qualified nurses, and calibrated blood pressure machines are used. We have included this information in the manuscript (lines 90-91). These study procedures minimise misclassification and measurement bias. Again, we have made this clearer in the manuscript. As indicated in our manuscript, this definition has been used in other analyses of HIV cohorts in SSA [2–4] and a prior analysis in the ALT cohort [5]. Again, we clarify that this definition is also consistent with international Hypertension diagnosis and management guidelines that recommend two consecutive abnormal blood pressure measurements[1]. 

2. General comment 2: In addition, since the cohort does not appear to be systematically selected, I'm not entirely sure about some of the conclusions. For instance, age has an association with hypertension, but I don't know if that's true or an artifact of the selection.

Response: We are unsure we understand what the reviewer means by saying that this cohort is "not systematically selected". However, to clarify further, we have stated that this cohort was established in 2014, and the inclusion criteria for the cohort are clearly stated (lines 81-84). We have also indicated that data is collected in accordance with the study protocol. The cohort is also registered with clinical trials.gov (lines 81-84). We also cited a paper that clearly defines the set-up of this cohort and its objectives (line 82). Specifically, we have clarified that "The analysis was conducted within the antiretroviral treatment long-term (ALT) cohort (ClinicalTrials.gov #: NCT02514707), which has previously been described [17,18]. Briefly, the ALT study is an ongoing single-centre prospective cohort of 1000 PLWH on ART for at least 10 years at baseline. Cohort participants were recruited from the Infectious Diseases Institute (IDI) HIV clinic in Kampala, Uganda, between 2014 and 2015.". 

Regarding age, we are not aware of any literature that has disputed age as a risk factor for hypertension. Ageing is one of the strongest risk factors for hypertension [6], and the evidence dates several decades. Age is associated with atherosclerosis and arterial stiffening. The paper by McEniery et al, explains the relationship between ageing and cardiovascular changes [7]. Hypertension is a common complication of ageing, and we do not believe that the finding is an artefact or a chance finding. In the methods section (lines 137-138) we have clarified that variables considered in the analysis were carefully selected based on prior published literature linking them to hypertension. 

Specific comments:

1. Specific comment 1: (lines 123-132) I am sure I follow how you performed your variable selection here and have three comments about this section. First, stepwise variable selection procedures, which include backward selection, usually do not do a good job of finding the most appropriate model (e.g., https://doi.org/10.1002/sim.3943). Stepwise procedures and any p-value based selection have quite a bit of evidence suggesting that they are poor at selecting the appropriate variables. For a decent summary, see the link above. Second, Sun et al. (http://dx.doi.org/10.1016/0895-4356(96)00025-X) found that bivariate screening can miss a variable that may be a confounder even when a p-value is as high as what you have (0.25). Bivariate screening can be considered a form of stepwise variable selection, which usually do not do a good job of finding the most appropriate model (e.g., https://doi.org/10.1002/sim.3943). Third, I don't understand how AIC is involved the selection. You've already used stepwise procedures to select variables, so what are you comparing? Generally, it's better to perform the entire selection based on more robust criteria, especially measures which assess the fit of the model (such as AIC or BIC where you could you Burnham and Anderson (doi: 10.1177/0049124104268644) as a guide) or, better yet, a shrinkage-based estimator such as lasso or lars.

Response: We thank the reviewer for providing the references above. As the reviewer may already know, variable selection and model fitting is a heavily debated issues in statistical analysis. The reviewer well describes the problems of automated algorithms for model fitting, and this is well reflected In the paper by Wiegand (https://doi.org/10.1002/sim.3943) that the reviewer refers to. First, we would like to clarify that we did not use stepwise algorithms to fit the reported models since we agree that this algorithm (Stata's stepwise command) is problematic and usually erroneous[8]. We fitted our regression models manually, carefully considering both significant variables and confounders. Stata's stepwise command was not used to fit the models. The paper by Sun et al. (http://dx.doi.org/10.1016/0895-4356(96)00025-X) that the author has suggested highlights the problem of using an automated selection of variables at the bivariable level. The paper criticises the use of a lower-level threshold for variable selection, and the example in the paper uses a threshold of p=0.05 . The authors of the paper correctly state that "If the statistical p value of a risk factor in the bivariable analysis is greater than an arbitrary value (often p = 0.05), then this factor will not be allowed to compete for inclusion in multivariable analysis.). We agree that this threshold would miss important confounders and other variables with borderline significance at bivariate analysis. It is important to note that the confounding variables in the paper had a p-value <0.25. The p-values for these variables would not have been dropped at the bivariate stage. Finally, the use of p-value<0.25 has been also used in several analyses, including a hypertension prediction model[9]. In their paper Chowdhury and Turin[9], make a case for bivariate analysis and variable selection. "It has also been suggested that variable selection should start with the univariate analysis of each variable. Variables that show significance (p<0.25) in the univariate analysis, as well as those that are clinically important, should be included for multivariate analysis.6 Nevertheless, the univariate analysis ignores the fact that individual variables that are weakly associated with the outcome can contribute significantly when they are combined. This issue can be solved partially by setting a higher significance level to allow more variables to illustrate significance in the univariate analysis.". We also agree that it is important to include "variables that are clinically important". This purposeful selection of the variables was our approach and consistent with published literature, including in a publication "Hosmer DW, Lemeshow S , Sturdivant RX . Applied logistic regression. New York: John Wiley & Sons, Incorporated, 2013". 

Secondly, the aim of this analysis was to identify risk (or protective) factors that are causally related to hypertension (explanatory modelling) and not to find the combination of factors that best predicts a current diagnosis or future event of hypertension (predictive modelling). The argument for explanatory modelling is that variable selection helps determine the variables that are related to the outcome, and this makes the model complete and accurate. Second, it helps select the most parsimonious model by eliminating irrelevant variables that decrease the precision and increase the complexity of the model[10]. This approach also reduces the risk of over-adjustment and unnecessary estimates [11]. Nevertheless, we agree that confounders should not be dropped from the model even when they are non-significant. All variables dropped from the multivariate model were assessed to determine if they were confounders, in which case they were retained. Specifically, BMI and age have consistently been shown to be confounders for hypertension [24,29–32]; their confounding potential was checked, and the variables were consistently included in the model (lines 145-148). Therefore, our variable selection was purposeful, and we did not exclude any confounders from the adjusted model. 

In addition, based on the criticism for the staged approach for model fitting, we have dropped this approach. We carefully fitted the model manually and the steps for model fitting are described (lines 134-148). Therefore, we fitted our model manually and variable selection was based on the significance of the variables and published data linking the variables to hypertension. To check the robustness of the final model, the goodness-of-fit test was used to determine the fitness of the final models (versus the full mode, i.e. with all variables), and the models with the smallest Akaike information criterion were considered the best-fitting [8]. We agree with the observation by Greenland [8] that "Unfortunately, all model-selection methods are subject to error, and no optimal method for selecting the best model form is known." Nevertheless, we carefully selected the final model and conducted appropriate model diagnostic procedures. 

Finally, we would like to clarify that the variables considered in the multivariable model were based on published data that linked these variables to hypertension (lines 137-138). Variable selection based on knowledge is a common approach in epidemiological modelling [12].

We hope that the reviewers will find our clarifications helpful. 

2. Specific comment 2: (lines 133-137) First off, it is possible to test to see if missing data are missing completely at random (https://doi.org/10.2307/2290157). I would do that first and, if the data are MCAR, not worry about imputation. If the MCAR test fails, then I would promote using the MICE output above the complete case analyses. Finally, for the imputation model to work best, one really needs to include as much data as possible, i.e., include auxiliary data. If you have more data, I encourage you to include that here.

Response: We agree entirely with this suggestion by the reviewer. We have conducted the regular Little's MCAR test (in Stata) test [13] (lines 159-160), and the results are provided under the sensitivity analysis section (lines 240-244). The regular Little's MCAR test gave a χ2 distance of 42.86 with 32 degrees of freedom, P=0.095. The null hypothesis for Little's MCAR test is that the data are missing completely at random (MCAR). Therefore, we do not reject the null hypothesis and we conclude that data are missing completely at random. Therefore, as suggested by the reviewer, we have removed the results based on multiple imputations as we have evidence that data were missing at random. 

3. Specific comment 3: Is follow-up time included as an offset in the Poisson model? This is never explicitly stated in the methods, so I wanted to check to make sure this was done. If this was not done, this must be done; otherwise, you are not calculating IRRs.

Response: We agree with the reviewer that the standard approach for modelling rates using Poisson is to include the logarithm of person-years of follow-up in the model as an offset term to account for the observation time of individuals. This is exactly what we did, and we have made this clearer in the manuscript (see lines 129-131). 

4. Specific comment 4: (Table 2) Editorial note: it's hard to read your table with the variable names centred and no lines or breaks between the variable categories.

Response: We agree with this suggestion and have revised the tables as suggested. We hope the tables are more readable (see the revised tables). 

5. Specific comment 5: (Tables 2 and 4) Is the "global p" coming from a Wald type 3 test? I strongly encourage you to include more information on the methodology in the methods section or drop this altogether. If these are Wald type 3 tests, I will admit that I have always found the results from these confusing and I think your analyses here highlight some of those. For instance, in Table 2, for most binary variables in the adjusted model, the p-value of the non-reference group and global p are identical but not for Sex. For alcohol history, the two p-values are pretty high (0.945 and 0.702), but the global p is far smaller (0.182). Though, in a roughly similar situation for Baseline LDL, the global p is larger than all the individual comparisons. From my experience, most people will pay attention to the individual categories and the global tests are going to create confusion.

Response: We understand the preferences of the reviewer and the observation that the global p-values for categorical variables may be confusing and are often misunderstood. In line with the suggestion by the reviewer, we have dropped the global p-values from the tables altogether (See tables 2 and 4). 

6. Specific comment 6: (Tables 2,4) Instead of categorising the continuous variables in the models (e.g., age, BMI, GFR), could you analyse them as continuous, maybe with a spline or other transformation? If you are really interested in characterising the associations, that would provide more information than categories or assuming linearity (as was done with age in 5-year increments). The categorisations of these variables are not explained in the methods and if the categorisations don't have clinical or policy relevance, they may lack utility.

Response: We agree with the reviewer that modelling variables as continuous variables is the preferred approach because assumptions of linearity and dose-response can be assessed. However, modelling continuous variables assumes linear associations, which are not valid for many clinical parameters [14]. For example, the associations between BMI and hypertension may not be linear in some populations [15].

We agree that categorising variables intuitively to give them clinical meaning produces data that is more useful to users. All variables (except age) were categorised to assign them categories that have clinical meaning. The justification has been provided, and the necessary references made. For example, the BMI categories follow the WHO categories; blood pressure categories are based on the European Association for cardiology categorisation for blood pressure. The CD4 categories are also based on WHO categories for immunosuppression status. We have added these justifications to the methods section. We have revised the methods section to make this clearer (see lines 138-144). We are hesitant to transform continuous variables (except age) as an interpretation of estimates becomes less straightforward and again, clinical decision-making (which is the basis of our analysis) rarely depends on continuous data but on categories with known risk. For example, it is more useful for a doctor (and more so for the low-level health cadres who provide HIV care in Africa) to know the risk of hypertension is twice higher if cholesterol is above 2 mmol/L rather than saying that the risk increases by 5% for every mmol/L increase in cholesterol. Please note that age increases hypertension was modelled as a transformed continuous variable because there is strong evidence supporting the linearity between age and hypertension risk [16].

 

Reviewer #2: 

This study sought to determine the prevalence and incidence of hypertension and its relationship with exposure to specific antiretroviral therapy in a Ugandan cohort of PLWH with long durations of ART.

Minor comments

1. Minor comment 1: Define what is meant by "heavily treatment experience".

Response: We have clarified that this refers to participants with prolonged exposure to ART at baseline (>10 years) (see lines 84-86)

2. Minor comment 2: Multivariable not multivariate.

Response: This change has been made throughout the entire paper. 

3. Minor comment 3: The authors use rate ratios and risk ratios interchangeability yet these differ (Line 127).

Response: We have addressed and consistently used incidence rate ratio (IRR)

4. Minor comment 4: The interpretation of odds ratios as percentages is not correct. (e.g., odds of prevalent hypertension increase by 16%).

Response: We have made the suggested revision. 

5. Minor comment 5: Define contemporary ART regimens. Line 76.

Response: Revised: We have clarified that these refer to antiretroviral agents that are currently recommended for HIV treatment (i.e., modern ART regimens) see line 76-77. 

6. Minor comment 6: What was considered as prior exposure to antiretroviral therapy in the staged model building approach line 115.

Response: Prior exposure means any exposure (use of) to antiretroviral before baseline and this has been clarified (line 137)

7. Minor comment 7: Provide references for definitions and analysis endpoints for example diabetes mellitus, hepatitis B, and chronic kidney disease.

Response: References have been added (lines 115-120)

Major comments 

1. Major comment 1: Clarify why those with more than two follow-up visits were excluded for the analysis of prevalence.

Response: We did not exclude participants with more than two follow-up visits as suggested by the reviewer but rather excluded participants with fewer than two follow-up visits (lines 102 and 171). The justification for this approach has been provided in the manuscript and is based on the definitions of our endpoint (hypertension). The definition of hypertension requires at least two measurements for hypertension diagnosis (refer to our definitions of endpoints and justification for two measures in the manuscript (lines 107-114). As noted in other responses above (reviewer 1), the definition is consistent with international guidelines for the diagnosis and management of hypertension[1].

2. Major comment 2: Was adherence to antiretroviral therapy considered in the analysis? How would it affect the estimates?

Response: We thank the reviewer for this question. Our cohort is comprised of highly treatment-experienced people living with HIV (on ART for at least ten years at baseline). Data on adherence data spanning ten years is not available in most cohorts, including ours. Therefore, we think modelling adherence at baseline would not be meaningful since it may not reflect adherence over the 10 years of antiretroviral therapy. Secondly, our focus was to determine if prior exposure (qualitatively) to antiretroviral agents was associated with higher rates of hypertension. However, we have modelled viral load which is a direct indicator of adherence. Lastly, we are unaware of any data showing a plausible causal relationship between adherence to ART and hypertension risk. Adjusting for both adherence and drug exposure would be problematic and lead to over-adjustment bias or unnecessary adjustment[11]. Finally, and possibly due to the reasons above, adherence to antiretroviral has not traditionally been included in models for cardiovascular diseases [17,18]. 

3. Major comment 3: How was Cumulative exposure to first-line and second-line ART modelled

Response: We modelled prior exposure and not cumulative exposure. We only provided data on cumulative exposure to antiretrovirals (table 1) as this data may have other program implications (e.g., ART resistance). Cumulative exposure has not been found to be a good predictor of cardiovascular events and this was demonstrated for abacavir in several cohorts [19,20]. Those studies reported that recent or current exposure, but not cumulative exposure to abacvir, was associated with cardiovascular risk.

4. Major comment 4: For the logistic regression model how were the covariates chosen? 

Response: We have clarified that for both logistic and Poisson regressions, the variable selection was based on published data on risk factors for hypertension in HIV cohorts [4,21–24] (lines 137-138). The procedures for model fitting and selection (i.e., variable selection for inclusion in the model) have also been provided in the methods (lines 144-150). Specifically, the regression models were manually fitted based on regression diagnostic procedures (AIC) and goodness of fit. In addition, during model fitting, care was taken to ensure that confounders were included in the final adjusted model. 

5. Major comment 5: It seems counterintuitive to remove untreated viral drugs that were not independently associated with hypertension (lines 117 -119), yet the objective of this study is to find the association between antiretroviral drugs and hypertension.

Response: We do not clearly understand what the reviewer refers to as "untreated viral drugs". However, we would like to inform the reviewer that we have revised the analysis based on concerns regarding the staged approach for model fitting (that we had used). This way, variables were only dropped if they were non-significant at the bivariable analysis or if they were insignificant in the adjusted model. We note further that if exposure to an antiretroviral drug was a confounder that variable was included in the adjusted model, even if it was not significant (see lines 145-146) 

6. Major comment 6: What was the rationale for choosing the non-ART confounders listed in line 121 - 122?

Response: The variable selection was based on published data on risk factors for hypertension in HIV cohorts [4,21–24] and we have made this clear in the methods (lines 137-138).

7. Major comment 7: Stepwise variable selection modelling is problematic in many ways including: it yields confidence intervals for effects and predicted values that are falsely narrow (Altman and Andersen, 1989) and biased regression coefficients that need shrinkage (see Tibshirani, 1996).

Response: As discussed above (response to reviewer 1), we would like to clarify that we did not use stepwise algorithms to fit the reported models because we also agree that this method of model fitting is problematic and usually erroneous[8]. Automated stepwise algorithms (e.g. Stata's stepwise command) were not used to fit the models. We have dropped this approach based on the criticism of the staged approach for model fitting. We carefully fitted the model manually, and the steps for model fitting are described (lines 134-138). The criticism of the stepwise approach is that it often drops confounders that may be non-significant[25]. We avoided this by fitting our models manually, and all variables dropped were assessed to determine if they were confounders, in which case they were retained (see lines 145-148).

As the reviewer may already know, variable selection and model fitting is a heavily debated issues in statistical analysis. The aim of this analysis was to identify risk (or protective) factors that are causally related to hypertension (explanatory modelling) and not to find the combination of factors that best predicts a current diagnosis or future event of hypertension (predictive modelling). Variable selection helps determine the variables that are related to the outcome, and this makes the model complete and accurate. Second, it helps select the most parsimonious model by eliminating irrelevant variables that decrease the precision and increase the complexity of the model[10]. While we agree with the observation by Greenland [8] that "Unfortunately, all model-selection methods are subject to error, and no optimal method for selecting the best model form is known.", there is an increasing suggestion that the purposeful selection of variables (like we did) is perhaps a good approach [26]. Our variable selection was based on the significance of the variables and with considerations made for confounding variables. 

We hope the reviewer will find our clarifications helpful. 

8. Major comment 8: Line 135. What was the evidence for the assumption of MAR?

Response: We have conducted the regular Little's MCAR test (in Stata) test [13] (lines 159-160), and the results are provided under the sensitivity analysis section (lines 240-244). The regular Little's MCAR test gave a χ2 distance of 42.86 with 32 degrees of freedom, P=0.095. The null hypothesis for Little's MCAR test is that the data are missing completely at random (MCAR). Therefore, we do not reject the null hypothesis and we conclude that data are missing completely at random. Thus, as suggested by the reviewer, we have removed the results based on multiple imputations as we have evidence that data were missing at random. 

9. Major comment 9: Tamper the discussion and conclusions given that the blood pressure measurements were annual with a high possibility of measurement error and visit-visit variability.

Response: We have included as a limitation that visits were annual, and therefore the precise estimation of hypertension incidence date may not be accurate (lines 281-282). However, as indicated in our methods, we have noted that blood pressure measurements are standardised and consistent with international guidelines. This approach minimises measurement bias. Regarding variability in blood pressure, we have noted in the methods that blood measurement is standardised in the clinics, and that blood pressure machines are well calibrated (lines 90-91). This minimises measurement bias. We are not aware of any data that suggests blood pressure measurements significantly fluctuate during the day. Available data suggest that blood pressure may dip during the night [27], but nocturnal blood pressure measurement is not the standard practice for hypertension screening. All study clinics are conducted during the day. What is clear is that blood pressure increases with ageing, and our results support that assertion. 

10. Major comment 10: Rate ratios on their own cannot be compared across populations. Instead, use standardised rates for comparison of incidence rates in the current study with those in other populations.

Response: We agree entirely with this observation and have added a caveat that "direct comparisons with other cohorts cannot be made" (see lines 259-261). Most studies on hypertension do not often report standardised rates and therefore direct comparison would still not be possible even if we made these estimates. 

 

Reviewer #3: 

1. Comment 1: The authors wrote that "all variables in this analysis were fixed at baseline". This rules out the possibility to evaluate time-varying phenomena, such as changes in eGFR. Please, comment on this.

Response: We agree with the reviewer that time-updated variables would be more intuitive and would allow the testing of whether some of these factors mediate the effect between some exposures and hypertension. However, time-updated analysis for some variables calls for caution since these may lie on the causal path between the exposure variable and hypertension. For example, there is strong evidence linking the use of protease inhibitors and chronic kidney disease [28]. Therefore time-updated eGFR would potentially attenuate or even neutralise the effect we want to see since low-eGFR may be the causal link between PIs and hypertension risk. This adjustment would lead to "over-adjustment bias" [11] and would potentially underestimate the effects of antiretroviral exposure. However, modelling time-updated variables. This pathway may also be true for several variables, including obesity, and diabetes. In addition, due to HIV funding mechanisms in Africa, funding for routine laboratory measures is not currently available. In Uganda, the US President's Emergency Plan for AIDS Relief (PEPFAR) funds only viral load testing. Therefore, most patients do not have repeated measures even when they are supposed to have annual tests. However, a laboratory panel of tests is frequently run at baseline for all patients. Therefore, modelling in African cohorts tends to use baseline data. This is important if one intends to produce reproducible data that is based on programmatic practices. 

Despite our arguments above, we agree that follow-up data on variables is important in causal modelling. We have noted the missing prospective data as a limitation in the discussion (line 280-281)

2. Comment 2: The authors raised the issue of missing variables. The proportion of missing data is reported only for some, but not for all, clinical variables, e.g., alcohol consumption. From the analysis of Table 1, there is a huge variance for HR of some variables (e.g., alcohol consumption). Please, add the number of missing data for all the variables.

Response: In Table 1, we have provided data on the missingness of data for all variables. Data on alcohol use was complete, and there was no missing variable on this variable (see table 1). Regarding the variance in the estimates for alcohol, this is not due to missing data as we had data on alcohol use for all included participants. The high variance is probably due to the few number of participants who had never used alcohol (n=16, 1.7%), and this is probably making estimates less precise. 

3. Comment 3: The authors wrote: "Multivariate logistic regression was used to determine the factors associated with hypertension at baseline". Further information is needed regarding the list of variables included in the multivariate model.

Response: We have listed and justified the factors considered for multivariable regression (lines 134-137). 

4. Comment 4: In the legend for Table 2, the authors wrote: "Covariates sex, prior AIDS, hepatitis B infection status, smoking history, prior exposure to lopinavir (LPV), abacavir (ABC), efavirenz (EFV), didanosine (ddI), high-density lipoprotein, baseline year, and baseline and nadir CD4 counts were not significant in the univariate analysis (P>0.2) and were not considered in the multivariate analysis". However, (1) sex is significantly associated to HTN prevalence; this should be clarified. (2) Prior exposure to EFV was reported in the table. Please, better specify the criteria adopted to design Table 2.

Response: We thank the reviewer for identifying this clerical error. Sex was indeed significant and was included in the final multivariate model. We have revised this (line 201). We have also revised the error regarding prior exposure to EFV. 

5. Comment 5 The authors wrote that multicollinearity was checked and a VIF >5 was used as cut-off. Considering that the whole number of variables examined in the univariate and subsequent multivariate model is 26, how many variables were excluded for multicollinearity?

Response: "Prior exposure to PIs" was collinear with "Prior exposure to LPV" (VIF=6.9). However, these variables were not significant in the final model (even when included one at a time) and they were dropped from the final multivariate model. The adjusted estimates for these variables were obtained by adding the variables back into the adjusted model, one at a time. We did not find any evidence for multicollinearity between the variables in the final model (lines 231-233). Please also note that not all variables considered for multivariable adjustment were included in the final multivariable (adjusted model). 

 

References

1 Unger T, Borghi C, Charchar F, Khan NA, Poulter NR, Prabhakaran D, et al. 2020 International Society of Hypertension Global Hypertension Practice Guidelines. Hypertension 2020; 75:1334–1357.

2 Okello S, Kanyesigye M, Muyindike WR, Annex BH, Hunt PW, Haneuse S, et al. Incidence and predictors of hypertension in adults with HIV-initiating antiretroviral therapy in south-western Uganda. J Hypertens Published Online First: 2015. doi:10.1097/HJH.0000000000000657

3 Brennan AT, Jamieson L, Crowther NJ, Fox MP, George JA, Berry KM, et al. Prevalence, incidence, predictors, treatment, and control of hypertension among HIV-positive adults on antiretroviral treatment in public sector treatment programs in South Africa. PLoS One Published Online First: 2018. doi:10.1371/journal.pone.0204020

4 RodrõÂguez-ArbolõÂ E, Mwamelo K, Kalinjuma AV, Furrer H, Hatz C, Tanner M, et al. Incidence and risk factors for hypertension among HIV patients in rural Tanzania-A prospective cohort study. PLoS One Published Online First: 2017. doi:10.1371/journal.pone.0172089

5 Mubiru F, Castelnuovo B, Reynolds SJ, Kiragga A, Tibakabikoba H, Owarwo NC, et al. Comparison of different cardiovascular risk tools used in HIV patient cohorts in sub-Saharan Africa; do we need to include laboratory tests? PLoS One 2021; 16. doi:10.1371/JOURNAL.PONE.0243552

6 Buford TW. Hypertension and aging. Ageing Res Rev 2016; 26:96–111.

7 McEniery CM, Wilkinson IB, Avolio AP. Age, hypertension and arterial function. Clin Exp Pharmacol Physiol 2007; 34:665–671.

8 Greenland S. Modeling and variable selection in epidemiologic analysis. Am J Public Health 1989; 79:340–349.

9 Chowdhury MZI, Turin TC. Variable selection strategies and its importance in clinical prediction modelling. Fam Med Community Health 2020; 8:e000262.

10 Chowdhury MZI, Turin TC. Variable selection strategies and its importance in clinical prediction modelling. Fam Med Community Health 2020; 8:e000262.

11 Lu H, Cole SR, Platt RW, Schisterman EF. Revisiting Overadjustment Bias. Epidemiology 2021; 32:E22–E23.

12 Walter S, Tiemeier H. Variable selection: Current practice in epidemiological studies. Eur J Epidemiol 2009; 24:733–736.

13 Li C. Little's test of missing completely at random. Stata Journal 2013; 13:795–809.

14 Dudenbostel T, Oparil S. J Curve in Hypertension. Curr Cardiovasc Risk Rep 2012; 6:281–290.

15 Tesfaye F, Nawi NG, van Minh H, Byass P, Berhane Y, Bonita R, et al. Association between body mass index and blood pressure across three populations in Africa and Asia. Journal of Human Hypertension 2007 21:1 2006; 21:28–37.

16 Lei X, Sun X, Strauss J, Zhao Y, Yang G, Hu P, et al. Health outcomes and socio-economic status among the mid-aged and elderly in China: Evidence from the CHARLS national baseline data. J Econ Ageing 2014; 4:59–73.

17 Bozzette SA, Ake CF, Tam HK, Chang SW, Louis TA, Diego S. Cardiovascular and Cerebrovascular Events in Patients Treated for Human Immunodeficiency Virus Infection. https://doi.org/101056/NEJMoa022048 2003; 348:702–710.

18 Ryom L, Lundgren JD, El-Sadr W, Reiss P, Kirk O, Law M, et al. Cardiovascular disease and use of contemporary protease inhibitors: the D: A: D international prospective multicohort study. Lancet HIV 2018; 5:e291–e300.

19 Elion RA, Althoff KN, Zhang J, Moore RD, Gange SJ, Kitahata MM, et al. Recent abacavir use increases risk for Types 1 and 2 myocardial infarctions among adults with HIV. J Acquir Immune Defic Syndr 2018; 78:62.

20 Dorjee K, Choden T, Baxi SM, Steinmaus C, Reingold AL. Risk of cardiovascular disease associated with exposure to abacavir among individuals with HIV: A systematic review and meta-analyses of results from 17 epidemiologic studies. Int J Antimicrob Agents 2018; 52:541–553.

21 Okello S, Kanyesigye M, Muyindike WR, Annex BH, Hunt PW, Haneuse S, et al. Incidence and Predictors of Hypertension in Adults with HIV Initiating Antiretroviral Therapy in Southwestern Uganda. J Hypertens 2015; 33:2039.

22 Okello S, Ueda P, Kanyesigye M, Byaruhanga E, Kiyimba A, Amanyire G, et al. Association between HIV and blood pressure in adults and role of body weight as a mediator: Cross-sectional study in Uganda. J Clin Hypertens 2017; 19:1181–1191.

23 Olack B, Wabwire-Mangen F, Smeeth L, Montgomery JM, Kiwanuka N, Breiman RF. Risk factors of hypertension among adults aged 35-64 years living in an urban slum Nairobi, Kenya. BMC Public Health 2015; 15:1–9.

24 Hatleberg CI, Ryom L, d'Arminio Monforte A, Fontas E, Reiss P, Kirk O, et al. Association between exposure to antiretroviral drugs and the incidence of hypertension in HIV-positive persons: the Data Collection on Adverse Events of Anti-HIV Drugs (D:A:D) study. HIV Med 2018; 19:605–618.

25 Sun GW, Shook TL, Kay GL. Inappropriate use of bivariable analysis to screen risk factors for use in multivariable analysis. J Clin Epidemiol 1996; 49:907–916.

26 Bursac Z, Gauss CH, Williams DK, Hosmer DW. Purposeful selection of variables in logistic regression. Source Code Biol Med 2008; 3:17.

27 Sherwood A, Steffen PR, Blumenthal JA, Kuhn C, Hinderliter AL. Nighttime blood pressure dipping: The role of the sympathetic nervous system. Am J Hypertens 2002; 15:111–118.

28 Ryom L, Dilling Lundgren J, Reiss P, Kirk O, Law M, Ross M, et al. Use of Contemporary Protease Inhibitors and Risk of Incident Chronic Kidney Disease in Persons with Human Immunodeficiency Virus: The Data Collection on Adverse Events of Anti-HIV Drugs (D:A:D) Study. Journal of Infectious Diseases Published Online First: 2019. doi:10.1093/infdis/jiz369

---

## [Decision Letter · Decision Letter 1]

16 Jan 2023

PONE-D-22-29381R1Prevalence and incidence of hypertension in a heavily treatment-experienced cohort of people living with HIV in UgandaPLOS ONE

Dear Dr. Byonanebye,

Thank you for submitting your manuscript to PLOS ONE. After careful consideration, we feel that it has merit but does not fully meet PLOS ONE’s publication criteria as it currently stands. Therefore, we invite you to submit a revised version of the manuscript that addresses the points raised during the review process.

We look forward to receiving your revised manuscript.

Kind regards,

Giuseppe Vittorio De Socio, MD, PhD

Academic Editor

PLOS ONE

Journal Requirements:

Reviewers' comments:

Reviewer's Responses to Questions

**Comments to the Author**

1. If the authors have adequately addressed your comments raised in a previous round of review and you feel that this manuscript is now acceptable for publication, you may indicate that here to bypass the “Comments to the Author” section, enter your conflict of interest statement in the “Confidential to Editor” section, and submit your "Accept" recommendation.

Reviewer #1: (No Response)

Reviewer #3: All comments have been addressed

2. Is the manuscript technically sound, and do the data support the conclusions?

Reviewer #1: Yes

Reviewer #3: (No Response)

3. Has the statistical analysis been performed appropriately and rigorously? 

Reviewer #1: Yes

Reviewer #3: (No Response)

4. Have the authors made all data underlying the findings in their manuscript fully available?

Reviewer #1: No

Reviewer #3: (No Response)

5. Is the manuscript presented in an intelligible fashion and written in standard English?

Reviewer #1: Yes

Reviewer #3: (No Response)

6. Review Comments to the Author

Reviewer #1: Thank you very much for your detailed explanations of your choices and adding clarifications that unfortunately I missed when I read this through the first time. I still have some disagreements on a few comments that I will list below, at least partially because of poor phrasing in my initial review and the manuscript.

1. As you have stated, you are exploring causality. That means you are relying on the sampling and the regression models to provide you with unbiased estimates of the causal effects. My "not systematically selected" comment was poorly phrased, but this is what I had in mind. Regression can do ok at providing unbiased estimates of causal effects in observational studies, but it's hard to know when regression does a good job without trying out other causal designs (e.g., propensity scores, disease risk scores, etc.). Thus, if your main goal is causality, it would seem to me to make sense to me to try one of these designs.

2. In that vein, I have a hard time with this statement in your response: "…the aim of this analysis was to identify risk (or protective) factors that are causally related to hypertension…". I don't see how you can do that without having balanced groups at baseline. That's why I brought up age. I'm not disputing that age is a risk factor. I mentioned age as an example because, if the age distribution is different between the hypertension and non-hypertension groups at baseline, that could cause bias. Currently you are relying on the regression model to remove that bias, which it may or may not be able to do. In table 1, the median age is ~2.5 years higher in the hypertensive group. Unfortunately, I'm not knowledgeable enough in this area to know if that is that enough to make a difference. It might be nothing, but this is where having some sort of balancing procedure might be useful. I'm also not able to know whether the regression model is able to remove any confounding that may have (in this case, I suspect the regression model is based on the info in table 1 since there is good overlap).

3. Also, if you are not concerned with "prediction", I suggest not using the term "predictors" in the abstract, the title of Table 4, and in the discussion. Again, maybe it's just me, but this is where some of my dissonance arises.

4. Thank you for your detailed explanation of your variable selection. I agree with your general thesis that the variable selection literature is a mess, especially in epidemiology. If one goes back far enough, it is possible to find any reference to support any claim. Unfortunately, I still see what you are doing as a form of sequential variable selection based on this statement from the review: "Therefore, we fitted our model manually and variable selection was based on the significance of the variables and published data linking the variables to hypertension" and lines 143-145. You may not be using a specific algorithm and you may not be using "stepwise" where variables can exit and enter the model, but you are still using the p-value (which I presume is what you mean by "significance") to select variables in a sequential manner. From what I have seen from simulation studies, using p-values for selection with sequential model fitting does poorly at capturing the true effect unless the sample size is really large. That said, it's hard for me to assess incorporating the change-in-estimate and AIC into the selection procedure. I don't know conditional on the p-value based selection whether those reduce confounding and, if so, by how much. If we are working in an 'anything goes so long as you explain it' situation regarding variable selection, then I think what you have is permissable. But, I am not really buying p-value based selection.

Reviewer #3: (No Response)

7. PLOS authors have the option to publish the peer review history of their article (what does this mean?). If published, this will include your full peer review and any attached files.

Reviewer #1: No

Reviewer #3: No

---

## [Author Response · Author response to Decision Letter 1]

24 Jan 2023

REVIEWER # 1

General comments:

1. General comment 1: Thank you very much for your detailed explanations of your choices and adding clarifications that unfortunately I missed when I read this through the first time. I still have some disagreements on a few comments that I will list below, at least partially because of poor phrasing in my initial review and the manuscript.

Response: We thank the reviewer for carefully reviewing our responses and the revised manuscript. I particularly thank the reviewer for the scientific propositions and stimulating arguments. In our responses below, we address the specific issues raised by the reviewer. 

2. Specific comment 1: As you have stated, you are exploring causality. That means you are relying on the sampling and the regression models to provide you with unbiased estimates of the causal effects. My "not systematically selected" comment was poorly phrased, but this is what I had in mind. Regression can do ok at providing unbiased estimates of causal effects in observational studies, but it's hard to know when regression does a good job without trying out other causal designs (e.g., propensity scores, disease risk scores, etc.). Thus, if your main goal is causality, it would seem to me to make sense to me to try one of these designs. 

Response: We understand the concern of the reviewer (see comment 3) is that we have used the term predictor, which in their view, suggests a causal relationship. We have made revisions to the manuscript and replaced "predictor" with factors associated". We understand the reviewer suggests that if we want to maintain the term "predictor," we should explore other statistical methods (e.g., propensity scores) as these, in their view, would better prove a causal relationship. Although propensity scores are increasingly used in statistical analyses, in our opinion, we do not agree that propensity scores are always superior to multivariate regressionIn a metanalysis of 43 observational studies Shah et al., demonstrated thatpropensity score methods give similar results to the conventional regression methods in observational studies [1]. In addition, Cepeda et al. demonstrated that the performance of propensity scores in large datasets with more than seven events per variable is similar to traditional multivariable regression methods [2] Therefore, we have maintained multivariate regression. However, we agree with the reviewer that the use of the term "predictors" may be strong in this case. To this end, we have revised the paper and dropped the use of "predictor" and used "factors associated".

3. Specific comment 2: In that vein, I have a hard time with this statement in your response: "…the aim of this analysis was to identify risk (or protective) factors that are causally related to hypertension…". I don't see how you can do that without having balanced groups at baseline. That's why I brought up age. I'm not disputing that age is a risk factor. I mentioned age as an example because, if the age distribution is different between the hypertension and non-hypertension groups at baseline, that could cause bias. Currently you are relying on the regression model to remove that bias, which it may or may not be able to do. In table 1, the median age is ~2.5 years higher in the hypertensive group. Unfortunately, I'm not knowledgeable enough in this area to know if that is that enough to make a difference. It might be nothing, but this is where having some sort of balancing procedure might be useful. I'm also not able to know whether the regression model is able to remove any confounding that may have (in this case, I suspect the regression model is based on the info in table 1 since there is good overlap).

Response: We thank the reviewer for this comment. We have addressed some of the issues in this comment above. As we understand age, the reviewer's concerns are that there are baseline differences between participants who developed hypertension and those who did not. The reviewer points out (as an example) a 2.5-year difference in median age and suggests that some balancing procedure should be done. We do not agree with this suggestion. We aim to determine whether these differences are associated with a higher risk of hypertension. Balancing (e.g., matching participants by age) would remove the effect of age, which we would like to show. If matching is done for all known predictors of hypertension, then the analysis will likely not show any predictors, which is our analysis goal. As argued above we do not think that traditional multivariate regression methods are inferior to propensity scores, especially if the sample size is not small as in our case. 

Specific comment 3: Also, if you are not concerned with "prediction", I suggest not using the term "predictors" in the abstract, the title of Table 4, and in the discussion. Again, maybe it's just me, but this is where some of my dissonance arises.

Response: We understand the genuine concern of the reviewer. We have revised the entire manuscript and dropped the use of "predictor" and used "factors associated". We hope the reviewer will find this change appropriate. 

Specific comment 4: Thank you for your detailed explanation of your variable selection. I agree with your general thesis that the variable selection literature is a mess, especially in epidemiology. If one goes back far enough, it is possible to find any reference to support any claim. Unfortunately, I still see what you are doing as a form of sequential variable selection based on this statement from the review: "Therefore, we fitted our model manually and variable selection was based on the significance of the variables and published data linking the variables to hypertension" and lines 143-145. You may not be using a specific algorithm and you may not be using "stepwise" where variables can exit and enter the model, but you are still using the p-value (which I presume is what you mean by "significance") to select variables in a sequential manner. From what I have seen from simulation studies, using p-values for selection with sequential model fitting does poorly at capturing the true effect unless the sample size is really large. That said, it's hard for me to assess incorporating the change-in-estimate and AIC into the selection procedure. I don't know conditional on the p-value based selection whether those reduce confounding and, if so, by how much. If we are working in an 'anything goes so long as you explain it' situation regarding variable selection, then I think what you have is permissible. But I am not really buying p-value based selection.

Response: We thank the reviewer for reading our responses on this issue. The reviewer is concerned by the "sequential variable selection based" and is not sure whether "conditional on the p-value based selection whether those reduce confounding and, if so, by how much". We agree with the reviewer that variable selection solely based on p-value would be inappropriate even when done manually. However, ask the reviewer to note that our variable selection was not exclusively based on p-value alone. For example, we have clarified that we tested whether variables were confounders (cause >10% change in effect size) before dropping them from the model (lines 145-149). Specifically, we included confounders in the final model even if their p-values were insignificant. P-value-based procedures would have dropped these, and we agree that the estimates would not be robust. In each case, we also tested whether the model with the "non-significant" confounder fits the data better than the model without the confounder. In that case, we used change-in-estimate and AIC into the selection procedure. We admit this was not clear in the manuscript, and we have revised the manuscript to make this clearer in the methods. Regarding our statement in the previous responses, "Therefore, we fitted our model manually and variable selection was based on the significance of the variables and published data linking the variables to hypertension", we intended to show that confounders were fitted in the final model even when they would have been dropped based on p-value (because of non-significant p-value). To further be sure of the estimates, we have rerun a full model (with all variables stated apriori, regardless of bivariate p-value), and the results are broadly similar. Also, the reduced final model is better fitting than a full model(AIC 849.693 vs 855.7153). Therefore, we have maintained the reduced final model as this is better fitting to the analysed data. 

Reviewer 3: No comments 

References

1 Shah BR, Laupacis A, Hux JE, Austin PC. Propensity score methods gave similar results to traditional regression modeling in observational studies: a systematic review. J Clin Epidemiol 2005; 58:550–559.

2 Cepeda MS, Boston R, Farrar JT, Strom BL. Comparison of logistic regression versus propensity score when the number of events is low and there are multiple confounders. Am J Epidemiol 2003; 158:280–287.

---

## [Editor Report · Decision Letter 2]

7 Feb 2023

Prevalence and incidence of hypertension in a heavily treatment-experienced cohort of people living with HIV in Uganda

PONE-D-22-29381R2

Dear Dr. Byonanebye,

We’re pleased to inform you that your manuscript has been judged scientifically suitable for publication and will be formally accepted for publication once it meets all outstanding technical requirements.

Kind regards,

Giuseppe Vittorio De Socio, MD, PhD

Academic Editor

PLOS ONE
---

## [Editor Report · Acceptance letter]

9 Feb 2023

PONE-D-22-29381R2 

Prevalence and incidence of hypertension in a heavily treatment-experienced cohort of people living with HIV in Uganda 

Dear Dr. Byonanebye:

I'm pleased to inform you that your manuscript has been deemed suitable for publication in PLOS ONE. Congratulations! Your manuscript is now with our production department. 

Kind regards, 

on behalf of

Dr. Giuseppe Vittorio De Socio 

Academic Editor

PLOS ONE